# Causes and Evolution of Winter Polynyas North of Greenland

Younjoo J. Lee[1], Wieslaw Maslowski[1], John J. Cassano[2,3], Jaclyn Clement Kinney[1], Anthony P. Craig[4], Samy Kamal[5], Robert Osinski[6], Mark W. Seefeldt[2,3], Julienne Stroeve[2,7], Hailong Wang[8]

[1]Naval Postgraduate School, Monterey, California, USA
[2]National Snow and Ice Data Center, Boulder, Cooperative Institute for Research in Environmental Sciences, University of Colorado, Boulder, Colorado, USA
[3]Department of Atmospheric and Oceanic Sciences, University of Colorado, Boulder, Colorado, USA
[4]Independent Researcher
[5]RedLine Performance Solutions, College Park, Maryland, USA
[6]Institute of Oceanology of the Polish Academy of Sciences, Sopot, Poland
[7]University of Manitoba, Winnipeg, Manitoba, Canada
[8]Pacific Northwest National Laboratory, Richland, Washington, USA

*Correspondence to*: Younjoo J. Lee (ylee1@nps.edu)

**Abstract.** During the 42-year period (1979–2020) of satellite measurements, four major winter (December–March) polynyas have been observed north of Greenland: one in December 1986 and three in the last decade, i.e., February of 2011, 2017, and 2018. The 2018 polynya was unparalleled in its magnitude and duration compared to the three previous events. Given the apparent recent increase in the occurrence of these extreme events, this study aims to examine their evolution and causality, in terms of forced versus natural variability. The limited weather station and remotely-sensed sea ice data are analyzed combining with output from the fully-coupled Regional Arctic System Model (RASM), including one hindcast and two ensemble simulations. We found that neither the accompanying anomalous warm surface air intrusion nor the ocean below had an impact (i.e., no significant ice melting) on the evolution of the observed winter open water episodes in the region. Instead, the extreme atmospheric wind forcing resulted in greater sea ice deformation and transport offshore, accounting for the majority of sea ice loss in all four polynyas. Our analysis suggests that strong southerly winds (i.e., northward wind with speeds greater than 10 m s$^{-1}$) blowing persistently over the study region for at least two days or more were required over the study region to mechanically redistribute some of the thickest Arctic sea ice out of the region and thus to create open water areas (i.e., a latent heat polynya). To assess the role of internal variability versus external forcing of such events, we carried out and examined results from the two RASM ensembles dynamically downscaled with output from the Community Earth System Model (CESM) Decadal Prediction Large Ensemble (DPLE) simulations. Out of 100 winters in each of the two ensembles (initialized 30 years apart one in December 1985 and another in December 2015), 17 and 16 winter polynyas were produced north of Greenland, respectively. The frequency of polynya occurrence had no apparent sensitivity to the initial sea ice thickness in the study area pointing to internal variability of atmospheric forcing as a dominant cause of winter polynyas north of Greenland. We assert that dynamical downscaling using a high-resolution regional climate

model offers a robust tool for process-level examination in space and time, synthesis with limited observations, and probabilistic forecasts of Arctic events, such as the ones being investigated here and
elsewhere.

## 1 Introduction

The Arctic has experienced amplified warming, both through the enhancement of global temperature rise, as well as through the reductions in sea ice and snow cover that impact the regional energy budget
(Serreze and Francis, 2006). This is commonly referred to as Arctic amplification (AA). On a seasonal basis, Arctic winter warming (AWW) exceeds summer warming by about a factor of four (Bintanja and van der Linden, 2013). In addition to sea ice variations, changes in atmospheric circulation have been linked to an increase in the frequency and duration of winter warming events in the Arctic (Graham et al., 2017). The trends in AWW and winter sea ice extent (SIE) have continued over the satellite record,
with the March SIE decline rate of ~2.6 % per decade for the period of 1979–2020, including the four lowest winter maxima of SIE in 2015–2018 and the lowest ten during 2005–2019. While SIE reductions in winter are less than ones in summer (Stroeve and Notz, 2018), there is clear supporting evidence that the winter ice pack has thinned and the amount of multi-year sea ice has declined (Kwok et al., 2009; Meier et al., 2014; Kwok, 2018; Ricker et al., 2021). It is expected that the resulting younger and
thinner ice is more susceptible to atmospheric wind forcing (Spreen et al., 2011; Itkin et al., 2017), yielding increased sea ice drift speed, enhanced fracturing, and more lead openings (Rampal et al., 2009). Hence, along with the current trend in the Arctic sea ice toward younger and thinner ice, it can be hypothesized that polynyas may become more prevalent in recent years. Since about half of the total atmosphere-ocean heat exchange over the Arctic Ocean may occur through leads and polynyas in winter
(Maykut, 1982), the occurrence and formation of polynyas could play a crucial role in the alteration of regional climate (Morales Maqueda et al., 2004).

In late February 2018, satellite imagery revealed an unusual open water polynya north of Greenland between the Lincoln and Wandel seas (Fig. 1a). This event received considerable attention not only
because it was claimed to be a one-of-a-kind extreme event involving some of the thickest Arctic sea ice but also because its emergence coincided with anomalous, above freezing, warming of surface air temperature over the region after the sudden stratospheric warming (SSW) event observed in mid-February 2018 (Moore et al., 2018). The presumed contribution of this warm surface air to the polynya opening is of interest and so are its causality, evolution, and past occurrences. Yet, detailed *in situ*
observations of polynyas are limited due to their intermittency and restricted access in winter. Hence, in addition to satellite measurements and weather data, fully coupled climate models become critical tools in studying such events (e.g., Ludwig et al., 2019). However, for the majority of global climate models (GCMs), many coastal polynyas are at sub-grid scales; thus, the GCM utility for comprehensive polynya studies is impeded (Weijer et al., 2017). In addition, GCMs are not intended to represent
specific climate events in space and time, and they are not suitable for process-level investigation of such events and quantification of their impact at both local and larger scales.

On the other hand, regional climate models (RCMs) used for dynamical downscaling are expected to show improvement in reproducing extreme weather events, given that their atmospheric boundary
conditions for simulations of the past to present are derived from global atmospheric reanalysis such as Climate Forecast System (CFS) version 2 (CFSv2; Saha et al., 2011). Therefore, high-resolution RCMs, such as the Regional Arctic System Model (RASM; e.g., Maslowski et al., 2012), offer unique capabilities for examining the spatio-temporal development and impact of observed specific events like a polynya (Fig. 1b), in the context of a fully-coupled climate system model (atmosphere-sea ice-ocean-
land), while CFSv2 is shown to be less skillful in simulating such an event (Fig. 1c). In addition, RCMs afford ensemble sizes prohibitive to their global fine-resolution counterparts, which is often a requirement to distinguish the forced response from internal model variability (e.g., Peings et al., 2021).

The 2018 winter polynya north of Greenland has been investigated by Moore et al. (2018) using an ice-
ocean model forced by surface atmospheric reanalysis. They have established the dominant role of surface winds in generating this polynya. Here, by taking advantage of the fully-coupled and high-resolution RASM hindcast simulation, combined with weather station and satellite sea ice data, we evaluate the capability of an RCM in reproducing the observed natural phenomena (Fig. 2) and investigate the coupled mechanisms involved in the development of northern Greenland coastal
polynyas within some of the thickest Arctic ice-pack cover. In particular, this study focuses on the analysis of historical winter (December–March) polynya events for the full period of satellite data availability (1979–2020) to diagnose the relative roles of thermodynamic and dynamic processes and to assess required forcing changes over the last four decades. Furthermore, by dynamic downscaling of the Community Earth System Model (CESM)-Decadal Prediction Large Ensemble (DPLE), two sets of
RASM ensemble (termed RASM-DPLE) simulations (30 years apart) are performed to examine the relative roles of internal variability and external forcing influencing the development of winter polynyas under two different sea ice regime scenarios: i.e., thicker ice in the 1980s versus thinner ice in the 2010s. We provide details of the satellite and weather station data used for this study in Sect. 2 and the model setup for the hindcast and DPLE simulations are described in Sect. 3. Next, Sect. 4 presents a
synthesis of observed and modeled past winter polynya results and examines the statistics of polynya occurrence and the required conditions for their generation in the RASM-DPLE ensemble simulations. This is followed by the discussion in Sect. 5 and the study is summarized in Sect. 6.

## 2 Data

### 2.1 Surface air temperature and wind

Hourly surface air temperature (3-hourly prior to 2015) data were obtained from the seven weather stations (World Meteorological Organization station identifier: 04221, 04254, 04285, 04351, 04330, 04312, and 04301) around Greenland (data available at https://rp5.ru) and then averaged daily for January–March of 2011, 2017, and 2018. Data are not available for the 1986/1987 winter. Since wind data are incomplete at the closest weather station from the center of the polynya in 2018 (i.e., Station

04301, Cape Morris Jessup at 83° 39' N and 33° 22' W), we used 3-hourly surface wind data from the adjacent Station 04312 (Station Nord at 81° 43' N and 17° 47' W), which are originally binned for 16 wind directions. Also, ERA-Interim atmospheric reanalysis of 6-hourly 10 m wind fields (Dee et al., 2011; https://www.ecmwf.int/en/forecasts/datasets/reanalysis-datasets/era-interim) was used for comparison with the RASM hindcast simulation over the study region.


## 2.2 Sea ice concentration and thickness

Daily sea ice concentration (SIC) data were obtained via the National Snow and Ice Data Center (NSIDC; https://nsidc.org/data/G02202/versions/4) for December 1978–March 2020 (Meier et al., 2021). Satellite-derived SIC used for this study is based on passive microwave measurements using the
NASA Team (NT) algorithm (Cavalieri et al., 1984); all the data are on a polar stereographic 25 km by 25 km grid. The daily SIC was used to examine the occurrence of a polynya in the satellite measurements. We later detected observed polynya events when the daily averaged satellite SIC dropped below 90 % over the study region. RASM sea ice thickness (SIT) was also compared with the CryoSat-2/the Soil Moisture Ocean Salinity (SMOS) satellite merged data that are only available in
winter months (Ricker et al., 2017) as well as CFSv2 reanalysis (https://doi.org/10.5065/D61C1TXF). Due to the lack of persistent SIT observations over the Arctic, the Pan-Arctic Ice Ocean Modeling and Assimilation System (PIOMAS) is often considered an "observational" proxy (Zhang and Rothrock, 2003; Schweiger et al., 2011; Stroeve et al., 2014). The PIOMAS (version 2.1) sea ice data were retrieved from the Polar Science Center at the University of Washington
(http://psc.apl.uw.edu/research/projects/arctic-sea-ice-volume-anomaly/data/).

## 3 Method

### 3.1 Regional Arctic System Model

RASM (see Maslowski et al., 2012; Roberts et al., 2015; DuVivier et al., 2016; Hamman et al., 2016;
Hamman et al., 2017; Cassano et al., 2017) is a limited-area, fully-coupled climate model, which consists of the Weather Research and Forecasting (WRF version 3.7.1) for atmosphere, the Los Alamos National Laboratory Sea Ice Model (CICE version 6.0.0) for sea ice and Parallel Ocean Program (POP version 2.1) for ocean, the Variable Infiltration Capacity (VIC version 4.0.6) land hydrology and routing scheme (RVIC version 1.0.0) for land. All the model components are coupled using the Craig et
al. (2012) version of the CESM flux coupler where the state values and fluxes are exchanged every 20 minutes of model time. RASM is configured over a pan-Arctic domain, including the entire Northern Hemisphere marine cryosphere and all terrestrial drainage basins that drain to the Arctic Ocean. The ocean and sea ice components share a horizontal resolution of 1/12-degree (~9 km) rotated spherical grid and are configured with 45 vertical levels for POP and five ice thickness categories for CICE. The
atmosphere and land hydrology components have a horizontal resolution of 50 km on a polar stereographic projection, with 40 vertical levels and a 50 hPa model top for the atmosphere, and three

soil layers for the land. RASM has been used for dynamical downscaling of data from coarse-resolution atmosphere since the benefit of RASM is in it being a fully-coupled model that has much higher resolution of ocean and sea ice processes than global climate models and atmospheric reanalyses. This

approach allows focused studies of the coupled Arctic system on seasonal to decadal timescales by allowing sensitive ice-ocean-atmosphere-land interactions across the coupled boundary layers to evolve more freely than an ice-ocean model forced with a data atmosphere. More details on the coupling of the atmospheric model to the coupler and other RASM component models, such as coupler to sea ice, are covered thoroughly in Cassano et al. (2017).


### 3.1.1 RASM hindcast simulation: September 1979–present

For RASM hindcast simulations, the initial, lateral boundary, and nudging conditions for WRF are created from reanalyses. In this study, CFS Reanalysis (CFSR; 1979–March 2011) and CFSv2 (April 2011–present) 6-hourly products are remapped to the RASM-WRF domain to provide the atmospheric

initial, lateral boundary, and nudging conditions. The CFSR global atmosphere resolution is approximately 38 km (T382) in the horizontal and 64 levels in the vertical (Saha et al., 2010). The CFSv2 atmospheric model has a global atmosphere resolution of approximately 100 km in the horizontal and 64 vertical levels. Nudging is applied to the top half of the RASM-WRF model domain, approximately above 540 hPa, to eliminate any potential drift of the interior of the domain towards

anomalous results and to maintain the large-scale features of the forcing data. The nudging of winds and temperature is limited to the top half of the model domain to allow the atmospheric model to evolve more freely to the model topography, WRF physics, and interactions with the fully-coupled model components in RASM. For RASM-POP, ocean temperature and salinity along the closed lateral boundaries are restored to monthly Polar Science Center Hydrographic Climatology version 3.0 (PHC

3.0; Steele et al., 2001). The initial conditions at the beginning of the hindcast simulation are derived from the 32-year spin-up of the ocean-sea ice model forced with the Common Ocean-Ice Reference Experiment Inter-Annual Forcing version 2 (CORE2-IAF; Large and Yeager, 2009) atmospheric reanalysis for 1948–1979.

### 3.1.2 RASM-DPLE ensemble simulations: initialized in 1985 and 2015

Analogous to the hindcast simulation, for the RASM-DPLE ensemble simulations, the initial, lateral boundary, and nudging conditions for WRM are created by remapping the global atmospheric output (6-hour intervals) from the initialized CESM-DPLE simulations, which are based on CESM, version 1.1 (Yeager et al., 2018). The CESM-DPLE atmosphere component is the Community Atmosphere Model, version 5 (CAM5; Hurrell et al., 2013), with a finite-volume dynamical core at nominal 1° horizontal

resolution and 30 vertical levels (Yeager et al., 2018). Each RASM-DPLE simulation is initialized with the ocean and sea ice conditions from the RASM hindcast simulation described above. Then, RASM-DPLE is integrated for 121 months with CESM-DPLE atmospheric forcing providing the lateral boundary conditions and data for the grid nudging of winds and temperature to the top half of the

RASM-WRF domain. RASM-DPLE output from the two 10-member decadal ensembles, initialized on
       1 December 1985 and 2015 with thinner ice in the latter period (Fig. S1), is selected for in-depth
       analysis under different regimes of the Arctic SIT distribution. The size of each ensemble (10 members)
       is determined by the availability of the required 6-hourly temporal resolution of the CESM-DPLE
       forcing data for the RASM-DPLE simulations. While similar atmospheric conditions are present in
terms of mean and variance among the members of each RASM-DPLE ensemble, the upper
       tropospheric condition between the two ensembles (1986–1995 vs 2016–2025) is found to be
       statistically different when analysis of variance (ANOVA) is performed for geopotential height at 300
       hPa in the central Arctic (Table S1 and Fig. S2). Hence, the 100 winters of RASM-DPLE output per
       each ensemble allows analysis of polynya occurrence under the past and near future climate.

**3.2 Self-organizing maps**

       Since atmospheric wind fields might be nonlinear in nature, a linear method, such as the empirical
       orthogonal function (EOF) analysis, may have drawbacks in extracting nonlinear information (Hsieh,
       2004). Hence, the self-organizing map (SOM), an artificial neural network based on a competitive
       learning algorithm (Kohonen, 2001), is used as an effective method for feature extraction of nonlinear
vector fields. The SOM has been widely used to visualize input data vectors onto a low-dimensional
       map of nodes and to objectively classify complex data sets in meteorology and oceanography (see Liu
       and Weisberg, 2011). For January–March 2018, the RASM/WRF pan-Arctic (>65 °N) 6-hourly surface
       wind fields (i.e., 10 m U and V components) are characterized using a SOM [4×4] map grid (i.e., 16
       nodes or patterns). Time-dependent spatial features of near-surface winds are identified and frequencies
of occurrence of wind patterns favorable for a polynya are quantified using the SOM Toolbox 2.0 for
       MATLAB available at http://www.cis.hut.fi/projects/somtoolbox.

       **4 Results**

       **4.1 2018 winter polynya – a case study**

       **4.1.1 Near-surface air temperature around Greenland**

Focusing on the most recent and the largest winter polynya event, daily near-surface air temperatures
       were examined from the weather stations around the Greenland coast for January–March 2018 (Fig. 3).
       A significant warming event (air temperatures rising above 0 °C) was observed over the northeastern
       Greenland region from mid-February to early March and captured well in the RASM simulation. This
       anomalous warming was most prominent along the northern Greenland coast (Figs. 3e and 3f), where
the polynya was observed and simulated in RASM (Figs. 2a and 2d, respectively), but less pronounced
       over the mid-eastern Greenland coast (Figs. 3g and 3h). In contrast, no warming was measured in the
       southwestern stations for the same period (Figs. 3b-3d). This anomalous warming, with relative
       humidity rising above 90 % (i.e., Station 04312; not shown), coincided with a strong reversal of the
       Arctic Oscillation (AO) index
(https://www.cpc.ncep.noaa.gov/products/precip/CWlink/daily_ao_index/ao.shtml) from a positive to a

strongly negative phase (Fig. 3a). Among the northeastern stations, the observed near-surface air temperature was positively correlated to the AO index (Fig. 3i). The maximum correlation coefficient (*r*) was time-lagged up to 11 days at the northern stations and only 3–4 days at the mid-eastern stations (Fig. 3j). The fact that the anomalous warming started a few days earlier at the mid-eastern stations and

ceased a few days later at the northern stations suggests the advection of warm air masses from the south. The prolonged warming at the northern stations could also be partly due to the release of oceanic heat from the polynya to the atmosphere.

On the other hand, near-surface air temperatures were inversely correlated with the AO index (Fig. 3i)

in the southwestern Greenland region, with a shorter time-lag (one to two days) for a maximum correlation coefficient (Fig. 3j). We found that no anomalous warming was present in the southwestern Greenland region during February. Near-surface air temperatures tended to gradually decrease from January through February, with the lowest temperatures observed between 22 and 24 February 2018 (Figs. 3b-3d). Thereafter, near-surface air temperatures rapidly increased at all southwestern stations,

peaking in early March, corresponding to the strong negative AO phase, and the second warming came approximately two to three weeks later.

Overall, the RASM hindcast simulation captured the sudden increase of near-surface air temperatures in northeastern Greenland and its gradual decrease remarkably well. RASM also reproduced well the

cooling in February and then the warming over southwestern Greenland, in terms of its magnitude and spatio-temporal variability (Fig. 3). One discrepancy in the RASM simulation against the weather stations data was a positive bias of near-surface air temperatures at Stations 04221 (Fig. 3b) and 04330 (Fig. 3g), possibly linked to the relatively coarse horizontal resolution of the RASM atmospheric component (i.e., 50 km), which is insufficient for resolving strong temperature gradients across the

ocean/land/ice sheet boundary or the fidelity of the near-surface temperature distribution over the Greenland Ice Sheet.

### 4.1.2 Sea ice dynamics

The sudden anomalous warming over northern Greenland with temperatures above 0 $^{o}$C was an extreme phenomenon in 2018, considering that the long-term (2011–2020) mean of February surface air

temperature is –27.8 $^{o}$C at Station 04301 (Fig. 3e; Cape Morris Jesup). In addition, there were other years of anomalous warming (Figs. 4b and 4c), albeit less pronounced, when previous, smaller, polynya events occurred during February 2011 and 2017 (Figs. 2b and 2c, respectively). RASM's realistic representation of the polynya, as well as the magnitude and timing of anomalous warming, grants confidence to the examination of the relative contribution of thermodynamic ice melt to the generation

of the polynya, although the center of the simulated polynya is not exactly collocated with the observed one. Figure 5 shows the thermal ice melting terms of RASM (at the surface, lateral, and bottom) were all negligible (< 1 cm) over the study region when integrated over February 2018. Hence, the dramatic rise of near-surface air temperatures by more than 25 $^{o}$C above climatology and their persistence around the freezing point for several days (Fig. 3e), had no impact on sea ice melt, nor on the preconditioning

and development of the polynya north of Greenland. In agreement with Moore et al. (2018), we

corroborate that this polynya was driven by mechanical redistribution of sea ice outside of the study region (see Fig. 4a); i.e., this was a latent heat polynya. Based on RASM results, we calculate that between 15 and 25 February 2018, 192 km$^3$ of sea ice was dynamically transported outside the study region (Fig. 4d). During the two weeks prior to the 2018 polynya event, mean thermodynamic ice

growth over the region was 0.72 km$^3$ d$^{-1}$ (Fig. 4d), which is comparable to the rate in a non-polynya year such as 2019 (not shown). The peak sea ice growth of 3.1 km$^3$ d$^{-1}$ occurred on 26 February right after the maximum daily dynamic ice removal and the anomalous warming period. However, this large sea ice removal during the polynya formation period was not fully replenished in the region by the end of March, even with dynamic and thermodynamic processes adding 81 km$^3$ and 42 km$^3$, respectively,

after 26 February. Overall, the RASM integrated thermodynamic ice growth in the study region during February 2018 was approximately 50 % higher (31.2 km$^3$; Fig. 4d) due to the rapid ice growth following the polynya opening, compared to the ice growth during a non-polynya year (i.e., 20.8 km$^3$ in February 2019; not shown).

### 4.1.3 Atmospheric-sea ice coupling

As dynamic processes dominated the overall winter sea ice in the study region, we have analyzed the wind data from the weather station (Station 04301; Station Nord) and have found that the polynya development was associated with strong and persistent winds from the south-southeast (Fig. 4g) in agreement with Moore et al. (2018) and Ludwig et al. (2019). We further examined how the spatial near-surface wind fields evolved over the time period associated with the sea ice divergence. The SOM

analysis extracted 16 patterns from a total of 360 synoptic wind fields from the 6-hourly RASM 10 m U- and V-wind components during January–March 2018. Figure 6 shows the four major wind patterns most frequently identified (more than 77 % occurrence) from the pre-polynya period (5 February 2018) until the closing of most of the open water areas (13 March 2018). The RASM hindcast simulation confirmed that this polynya event was predominantly associated with southerly to southeasterly winds

blowing over the northern Greenland region (Fig. 6b and 6c), consistent with the ERA-Interim reanalysis of 10 m wind fields (Fig. S3). Prior to the polynya event in 2018, surface winds were mainly from the north or northwest over the region (Fig. 6a), which yielded little or no sea ice divergence (Fig. 6e). When the major wind pattern shifted between southerly to southeasterly winds over the northern Greenland region on 15 February 2018 (Fig. 6b), sea ice started to deform and diverge significantly

(Fig. 6f). Beginning on 20 February 2018, the southeasterly wind became even more prominent, with 19 % stronger wind speed over the northern coast of Greenland (Fig. 6c), which increased the sea ice deformation rate further and led to the maximum polynya opening (Fig. 6g). The largest observed and modeled polynya areas were identified on 25 February 2018 (see Figs. 2a and 2d, respectively). Thereafter, within a week, the shift of wind patterns in late February (Fig. 6d) reversed the dynamic sea

ice volume (SIV) tendency from net loss to gain (Fig. 4d) and subsequently reduced the deformation rate back to nearly zero by early March (Fig. 6h).

## 4.2 Winter polynyas during the 2010s

### 4.2.1 February 2011 and 2017

Upon examining satellite-derived SIC over the data record between 2010 and 2020 (Figs. 7 and S4), we found that there were two additional polynya events in the same month, albeit smaller, over the northern Greenland region observed in February 2017 and 2011 (see Figs. 2b and 2c, respectively). Here, we found those polynya events when the daily mean satellite-derived SIC fell to or below 90 % over the region (see Figs. 7a, 7g, and 7h,). As was the case of the 2018 polynya, both the previous events also

coincided with anomalous warming, peaking on 12 February 2011 (Fig. 4b) and on 8 February 2017 (Fig. 4c), as measured at Station 04301 (Cape Morris Jesup). But, near-surface air temperature variability was not statistically correlated with the AO index (not shown). Those two polynyas were represented well in the RASM hindcast simulation (Figs. 2e and 2f) and thus we investigated their causality with respect to their relative SIV reductions due to dynamical processes and/or

thermodynamic ice melt (Figs. 4e and 4f). The RASM results confirmed that those were latent heat polynyas dominated by the mechanical redistribution of sea ice out of the region associated with southerly winds. Their sizes were smaller compared to the 2018 polynya, with the one in 2017 even smaller than the one in 2011, which we attribute to somewhat different wind patterns such as its direction, magnitude, and duration (Figs. 4i and 4h). Table 1 shows that during the polynya periods

based on the RASM simulation (defined here when ice volume tendency is less than $-10$ km$^3$ d$^{-1}$ for more than three days consecutively), i.e., 12–15 February 2011 and 8–10 February 2017, 55 km$^3$ and 42 km$^3$ of sea ice was dynamically removed outside the study region, respectively, which is much less compared to the ice loss of 189 km$^3$ during the 2018 event (16–25 February 2018). The size of a polynya was proportional to the pseudo-wind stress (i.e., wind speed squared) integrated over the

polynya period (Table 1). Thus, more turbulent (latent plus sensible) heat was lost during the 2011 winter polynya (daily mean of $-91.6$ W m$^{-2}$ and maximum of $-116$ W m$^{-2}$) when its size was bigger (Table 2). In addition, referenced to the February ice growth during a non-polynya year (for example, 20.8 km$^3$ in February 2019; not shown), the RASM thermodynamic ice growth integrated over the month of February was elevated: 33 % higher in 2011 (27.6 km$^3$; Fig. 4f) in the study region. Note that

the daily mean turbulent heat flux in the study region was $-60.9$ W m$^{-2}$ during the 2018 winter polynya event with the maximum daily heat loss up to $-124$ W m$^{-2}$. When integrated, the total turbulent heat loss in 2018 was one order of magnitude larger than in 2011 due to the size and duration of open water areas (Table 2).

### 4.2.2 What is driving changes in polynya frequency

Given the above analysis, an outstanding question is why winter polynyas became more frequently observed during the 2010s within the past four decades. Since observational data around northern Greenland are incomplete, we expanded the analysis of surface wind fields (10 m U- and V-components) from the RASM hindcast simulation near Station 04312 (Fig. 8a; Station Nord) and Station 04301 (Fig. 8b; Cape Morris Jesup). Our analysis revealed that northward wind (blowing from

the south; required for opening of a winter polynya along the coast of northern Greenland) has recently

become more frequent, stronger (i.e., wind speed >10 m s$^{-1}$), and more persistent (i.e., blowing for at least two consecutive days or longer). In addition, the three years of winter polynya occurrence satisfied the above criteria. Based on the RASM hindcast simulation, the mean wind conditions in February 2009 were similar to the conditions in February 2017 near Station 04312 (Fig. 8a), but a notable polynya was not detected, possibly owing to the influence of such wind conditions over a smaller area within the main polynya region. The observational data also indicate polynya-favorable wind conditions in 2009: i.e., relatively warmer air (–12.6 °C) blowing from the south-southwest with the maximum wind speed of 19 m s$^{-1}$ on 7 February 2009 at Station 04312 (Table S2). But, the satellite-derived mean daily SIC only dropped down to 94 % in February 2009 (Fig. S5o) because the wind was possibly weaker, compared to the wind conditions in February 2017 (Fig. 4i). Note that the early February 2009 data at Station 04312 are missing and the entire February 2009 data at Station 04301 are completely unavailable. Although changes in wind (i.e., direction and intensity) play a role in this region, we cannot completely rule out that a thinning of sea ice may promote more frequent polynyas in recent years. Hence, we additionally simulated two large ensembles at two different time scales, which span thicker and thinner regimes, and evaluated model representation of polynyas in Sect. 4.4.

## 4.3 Winter polynyas between the 1980s and 2000s

It is interesting to point out that no major winter polynya was found during the 1980s to 2000s (Figs. S4 and S5) except for one major event in December 1986; the lowest daily mean SIC was 82 % over the study region on 15 December 1986 (Fig. S5i). There were two instances indicating possible polynya events because the regional daily mean SIC was below 90 %, i.e., December 1984–January 1985 (Fig. S5g) and December 2002–January 2003 (Fig. S4i). However, they were not considered polynyas in this study since sea ice dynamic volume tendency (DVT) was not below the threshold ($\leq$ –10 km$^3$ d$^{-1}$) for at least three consecutive days based on the RASM simulation (Fig. S6). Compared to the recent polynyas in the 2010s, the polynya in December 1986 was larger than the ones in 2011 and 2017 but smaller than one in 2018. The satellite SIC revealed that the polynya occurred in a similar location to the ones in the 2010s (Fig. 9a). The RASM simulation reasonably captured it well (Fig. 9b) and confirmed that the December 1986 event was not associated with significant thermal ice melting (Fig. 9c) although it was slightly elevated, compared to the 2018 event (Fig. 5). Analogous to the recent 2010s polynyas, the December 1986 polynya was linked to a strong southerly wind of almost the same strength as the 2018 winter polynya (Fig. 9d and Table 2), but its duration was shorter (the strong southerly wind only lasted five days; Fig. S7).

Another feature in common is that anomalous atmospheric warming occurred during the polynya event (Fig. 9e). However, no SSW was reported during the winter of 1986–1987 (see Butler et al., 2017) and it was an El Niño winter in contrast to the polynyas of the 2010s. The RASM hindcast simulation suggests that between 12 and 16 December 1986, 142 km$^3$ of sea ice was dynamically transported outside the study region (Fig. 9f), which is 25 % less than the amount of ice removed in 2018, and this was 2.6 times more than the ice transported in February 2011 (Table 1). However, the daily mean turbulent heat flux was much less in December 1986 than in February 2011 (Table 2) even though the

polynya size was larger and the wind was stronger in December 1986. This is possibly due to the fact that sea ice was thicker in 1986 compared to recent years (see Table 1).

385

### 4.4 Polynyas in a large ensemble of initialized decadal prediction simulations

According to the merged CryoSat-2/SMOS data, the mean SIT over the region does not exhibit a negative trend in February. For example, in the first week of February, the mean SIT was 2.74 m in 390 2011, 3.16 m in 2017, and 2.73 m in 2018. On the other hand, the RASM hindcast simulation indicates a gradual thinning of sea ice in the region and a long-term trend of SIT was –0.26 m per decade for all months during 1985–2019 (not shown). In order to further investigate the potential role of regional sea ice thickness reduction versus internal variability in the occurrence of winter polynya events north of Greenland, we performed and examined the two RASM 10-member ensembles forced with atmospheric 395 output from the CESM-DPLE simulations. They are initialized 30 years apart, i.e., in December 1985 and 2015, respectively, to represent different SIT conditions over the study region, with the former corresponding to a thicker ice regime (Fig. S1a; mean SIT of 3.3 m) and the latter to a thinner ice regime (Fig. S1b; mean SIT of 2.3 m). Note that the PIOMASS SIT also corroborates that sea ice was 1.1 m thicker north of Greenland in November 1985 (Fig. S1d; mean SIT of 3.3 m) than in November 400 2015 (Fig. S1e; mean SIT of 2.2 m). Each ensemble member is integrated forward for 10 years, thus resulting in 100 winters (i.e., 10 ensemble members for each 10-year simulation) for statistical analysis of polynya occurrence in each of the two ensembles. Note that because of the setup of these experiments (i.e., dynamic downscaling of the CESM DPLE atmospheric output as boundary conditions for forcing the RASM-DPLE fully-coupled simulations), we only compared the probability of polynya 405 occurrence during winter months (December–March) instead of focusing on accurate temporal representation of such events in a similar location.

Taking the winter polynya in February 2017 (Fig. 4e and Table 1) as the baseline, which was the smallest among the four observed events, we defined "a latent heat polynya" in the RASM-DPLE 410 ensemble members when a daily winter sea ice loss due to dynamic processes was greater than 10 $km^3$ $d^{-1}$ for at least three consecutive days over the study area (see Fig. 4a). Note that the three other observed polynya events experienced 4, 5, and 10 consecutive days of dynamic sea ice loss greater than 10 $km^3$ $d^{-1}$ during February 2011 (Fig. 4f), December 1986 (Fig. 9f) and February 2018 (Fig. 4d), respectively (Table 1). Analogous to the RASM hindcast simulation, Tables 3 and 4 list all the potential 415 polynya occurrences from the two RASM-DPLE ensembles based on the volume outflow. During these events, the polynya-favorable winds were similar between the two ensembles in terms of mean wind speed and stress as well as their variance. Those wind data sets, assumed to be non-parametric, were compared using the Kolmogorov-Smirnov (K-S) two-sample test which failed to reject the null hypothesis with a 5 % significance level (not shown). The total ice volume removal was highly 420 correlated with the wind stress ($r=0.78$ and 0.76 in Tables 3 and 4, respectively) but not with the initial ice thickness where the K-S test rejected the null hypothesis with a 5 % significance between the two ensembles (not shown). This suggests that the ice removal in winter polynyas was dependent on the wind condition rather than the initial ice condition.

Considering that sea ice was thicker in the 1985 RASM-DPLE ensemble, an additional threshold was applied to define a winter polynya: at least 10.8 % of the total ice volume removal referenced to the initial sea ice similar to the one in February 2017 (Table 1). Once factoring in the initial ice thickness, we found 17 polynyas in the 1985-initialized ensemble and 16 polynyas in the 2015-initialized ensemble (see Tables 3 and 4, respectively), out of the 100 winters each. Note that some ensemble members simulated more than one polynya event while some had no polynya at all. When we used a higher threshold of relative ice volume removal like the February 2011 event (i.e., 13.0 %), the same number of winter polynyas was detected: 14 polynyas in both of the RASM-DPLE ensembles. The longer-lasting and larger polynyas (with dynamic sea ice removal $\geq 10$ km$^3$ d$^{-1}$ for five consecutive days or longer) were also similarly produced in the RASM-DPLE ensembles: four incidents in the 1985-initialized and five incidents in the 2015-initialized runs with 21.9-35.1 % relative sea ice volume loss.

We finish this analysis of RASM-DPLE results by comparing the longest-lasting polynya detected in each ensemble (see Tables 3 and 4), i.e., from the ensemble member #4 in January 1988 (Fig. 10a) and the ensemble member #2 in January 2024 (Fig. 10b). As in the case of observed events, these latent heat polynyas were created due to dynamic sea ice transport (DVT $\leq -10$ km$^3$ d$^{-1}$ for seven consecutive days), resulting in significant sea ice removal of $-120$ km$^3$ out of the region in 1988 (Fig. 10c; Table 3) and $-136$ km$^3$ in 2024 (Fig. 10d; Table 4). By cross-examining wind patterns over northern Greenland (i.e., near Cape Morris Jesup), our analysis confirmed that these large polynya openings were also associated with very strong and persistent southerly winds in the RASM-DPLE simulations (Figs. 10e and 10f). We found that daily DVT and meridional wind were significantly correlated, and the size of polynyas was highly dependent on the pseudo-wind stress integrated over the polynya period (Tables 3 and 4). However, given the wind patterns, their duration, and the integrated sea ice removal ($-189$ km$^3$) during the observed polynya in February 2018 (Table 1), this event stands out within all the RASM results analyzed in this study. Compared to the 2018 event, the slight difference in the magnitudes between the largest polynyas of each ensemble still does not imply much significance of SIT in their generation and evolution. Overall, the similar number of winter polynyas, produced in the two RASM-DPLE ensembles 30 years apart, implies that thinning of sea ice is not a significant contributor (at least up to now) to the generation of such events for this region during wintertime.

## 5 Discussion

Recent studies (i.e., Moore et al., 2018; Ludwig et al., 2019) used an ice-ocean model to study the polynya in February 2018, which means that the model is one-way coupled. Although net heat fluxes are calculated at the surface in the forced ice-ocean model, they are not exchanged across the air-sea interface. On the other hand, RASM, a fully-coupled regional Earth system model, is used for dynamical downscaling, and thus a lower atmosphere, including the surface conditions, is predicted every coupling time step by exchange of fluxes and state variables with the ocean, sea ice, and land components. Although the RASM near-surface wind fields agree well with the reanalysis data, it is possible that a slight discrepancy in wind direction or magnitude near the study region may have shifted

the center of the polynya more westward than was observed in February 2018, which can be attributed to the relatively coarse resolution of the atmosphere (50 km WRF). This suggests that the RASM-WRF

resolution needs to be further improved for better simulating coastal polynyas. Although the RASM hindcast simulation relies on the CFSv2 atmospheric conditions, the winter SIT on 25 February 2018 is very unrealistic in CFSv2 with too thick ice in the central Arctic (Fig. 1c). This is a common bias in current climate models (i.e., Watts et al., 2021), resulting from limitations in or lack of representation of key physical processes, partly due to their coarse model resolution (e.g., Chassignet et al., 2020). Since

the nudging of CFSv2 winds and temperature in WRF are used only above approximately 540 hPa in the RASM hindcast simulation, the dynamic downscaling approach using the high-resolution fully-coupled model allows resolving the fine-scale processes and provides valuable insight into the mechanism of the generation and evolution of winter polynyas off the northern coast of Greenland. We examined the occurrence of such events in RASM against the satellite observations in winter months

(December–March) over the past four decades (1979–2020). The results from the RASM hindcast simulation suggest that the size of the winter latent heat polynyas observed in this region is sensitive to the direction, magnitude, and duration of near-surface winds.

Subject to the limited sample sizes, we find that the generation of a winter polynya in this region
requires strong southerly winds (i.e., speeds >10 m s$^{-1}$ and lasting for more than two days largely over the study region based on the RASM hindcast simulation). Table 1 and Figure 8 show that the stronger and more persistent the southerly wind blows, the larger the winter polynya becomes regardless of ice conditions in the study region, but it is not immediately clear what causes such changes in wind patterns. At the same time, it is reported that the southerly winds might reduce sea ice export through

Fram Strait by a slowdown of sea ice drift (Wang et al., 2021). The RASM simulation captures such decline in 2018 and overall agrees with the observed interannual variability of ice export through the Fram Strait (Fig. S8; see also Smesdrud et al., 2017), indicating that atmospheric wind variability is well represented over the region. However, apart from the polynya region, sea ice coverage is overestimated especially north of Svalbard (Fig. 1b) where basal melting plays a role (Fig. 5c). This

may suggest that ocean heat delivered via the West Spitsbergen Current is not strong enough to melt sea ice along its path into the Arctic. It can also be speculated that this region might have been affected by increasing extreme winter storm activity in recent years that is associated with anomalous warming events. However, no significant trends were found in January–February during 1979–2015 (Rinke et al., 2017).


Considering the overall vulnerability and fate of some of the thickest sea ice in the Arctic under the recent warming climate, sea ice may become susceptible to modulation by atmospheric forcing. Sensitivity experiments indicate that a polynya would occur under the same atmospheric conditions of February 2018 even in the thick ice conditions like the winter of 1979 (Moore et al., 2018).

Using RASM, the combination of decadal dynamical downscaling with an ensemble approach allows us to increase the sample size to 100 winters for each of the two ensembles and thus quantitatively evaluate the impact of decreasing ice thickness on the occurrence of polynyas in the region. The RASM-DPLE study shows that the frequency, size, and duration of winter polynyas are unaffected by sea ice thinning, showing no apparent difference in polynya size or frequency of occurrence over time between the two

ensembles initialized with two different sea ice regimes 30 years apart (1980s vs 2010s). One interesting feature is that winter polynyas mostly (65 %) occur in December–January in the 1985-initialized simulations whereas they are more prevalent (75 %) in January–February in the 2015-initialized ensemble runs. In addition, the range of 5-day mean SIT before the occurrence of each polynya was 2.9–4.4 m (with a mean of 3.7 m and standard deviation of 0.36 m) in the 1985-initialized RASM-DPLE ensemble and 1.8–3.4 m (with a mean of 2.8 m and standard deviation of 0.36 m) in the 2015-initialized RASM-DPLE ensemble (Table 3 and 4, respectively). Although sea ice is thicker, the 1985-initialized ensemble simulations produced only one more polynya than the 2015-initialized ones. Hence, regardless of the SIT decline over the region, this study suggests that the primary necessary condition for a winter polynya occurrence is strong and persistent southerly winds. With the maximum wind speed exceeding 20 m s$^{-1}$ and a duration of 10 days, the 2018 winter polynya remains unique in the context of any of these metrics.

When the polynyas occurred over the region, they coincided with a reversal (from positive to negative) of the daily AO index. For instance, during the 2017–2018 winter, the AO shifted significantly from a positive phase to a strong negative phase between mid-February to early March (Fig. 3a), which was associated with the weakening of the polar vortex and allowed warmer air into the Arctic. But, the relationship between the AO index and near-surface temperature north of Greenland was not statically significant during the 2011 and 2017 winter polynya months. Alternatively, Kim et al. (2014) argued that due to sea ice loss especially in the Barents-Kara seas, the weakened stratospheric polar vortex preferentially induced a negative phase of the AO at the surface, resulting in warm air moving into the Arctic. This anomalous warming event occurred coincidently after the SSW event observed in mid-February 2018 (Moore et al., 2018; Rao et al., 2018), which developed into a vortex split (Lü et al., 2020). Subsequently, the winter weather was severe with intense cold air across Europe in March 2018 (Overland et al., 2020). Although the exact cause of SSW variability is still under debate, SSWs are generally known to cause anomalous warming over Greenland and impact surface weather patterns down to mid-latitudes (Butler et al., 2017). The frequency of their occurrence is enhanced by El Niño conditions (Polvani et al., 2017) and may also be enhanced during La Niña winters depending on SSW definitions (Song and Son, 2018). For example, La Niña was in the winter of 2017–2018, while El Niño was in December 1986. In the recent winters with SSW (Butler et al., 2017; Rao et al., 2018; Knight et al., 2021), no winter polynya events occurred except in February 2018. In other words, the winter polynyas in 1986, 2011, and 2017 were not associated with SSW.

The opening of polynyas in the region between the Wandel and Lincoln seas primarily depends on a large-scale surface pressure pattern change, resulting in the strengthening of intense southerly winds that are short-lived and sporadic. Toward the end of each polynya event, the rate of thermodynamic ice growth generally increased (Figs. 4d, 4e, and 4f). The maximum rate of ice growth happened not when sea ice removal was largest with the strong southerly wind but when air temperature started to significantly drop (Fig. 4). Upward surface turbulent heat fluxes continued even after DVT became positive (i.e., Table 2 and Fig. 4) when the wind direction was reversed (i.e., Fig. 6d) until the open water area was completely covered by ice. In general, the rate of new ice formation in winter is expected to be high during an open water phase, such as leads and polynyas, due to intense turbulent

heat (sensible and latent) loss to the atmosphere. In some polynya regions, turbulent heat loss is as high as 300 W m$^{-2}$ such as along the Weddell Sea coast (see Morales Maqueda et al., 2004). However, the mean daily turbulent heat loss in the study region during the polynya in 2018 was about 60.9 W m$^{-2}$
(with a maximum of 124 W m$^{-2}$), which is in good agreement with the results of Ludwig et al. (2019) based on the forced ice-ocean model NAOSIM (mean and maximum heat fluxes of 40 and 124 W m$^{-2}$, respectively). Those values are much smaller than that in the St. Lawrence Island polynya (412 W m$^{-2}$; Pease, 1987) and Okhotsk Sea coast (471 W m$^{-2}$; Alfultis and Martin, 1987), but comparable to the Northeast Water (NEW) polynya (31 W m$^{-2}$; Morales Maqueda et al., 2004). Nevertheless, new sea ice
formation was somewhat slowed down in the study region because the air-sea temperature difference was much smaller than in the other regions listed above due to the anomalously warm air carried by southerly winds over northern Greenland.

RASM estimated a maximum winter polynya size of 13,000 km$^2$, where SIT is less than 10 cm, in 2018
while Ludwig et al. (2019) calculated it as large as approximately 60,000 km$^2$ using satellite SIC, although there are large uncertainties between different satellite algorithms. For example, MODIS SIC underestimates open water areas by about 50%, compared to the Advanced Microwave Scanning Radiometer 2 (AMSR2) SIC (see Fig. 3 in Ludwig et al. (2019)). On the other hand, the RASM polynya size is based on SIT and its maximum size could increase to 29,400 km$^2$ if a threshold of SIT less than
25 cm is applied. According to the RASM simulation between 1 February to 31 March 2018, the study region lost approximately 208 km$^3$ of sea ice due to mechanical ice removal. Although sea ice was added later by the dynamic replenishment (102 km$^3$) and the thermodynamic ice growth (64 km$^3$), the deficit of SIV was about 42 km$^3$, which is equivalent to the 36 cm reduction of mean SIT over the study region. However, von Albedyll et al. (2021) showed that the thermodynamic and dynamic processes
almost equally contributed to the mean SIT change between 25 February and 31 March 2018. Ludwig et al. (2019) calculated the ice volume thermodynamically produced (33 km$^3$) including the polynya period (14 February to 31 March 2018) using the freezing degree day parameterization, but RASM produced slightly more sea ice for the same period (53 km$^3$). Although an ice cover was re-established with a mean thickness of 1.96 m at the end of March 2018 (von Albedyll et al., 2021), sea ice in the
northern Greenland region was not fully restored even a month after the peak ice removal on 25 February 2018 unlike the events in 2011 and 2017 (Fig. 4d). The CryoSat-2/SMOS data also confirmed the similar sea ice loss in the region; at the end of March 2018, the mean sea ice north of Greenland was relatively thinner (2.45 m), compared to the beginning of February (2.73 m). Therefore, one could hypothesize that the 2018 winter polynya event could have contributed to the preconditioning of the
polynya event in the following summer (Schweiger et al., 2021), which was observed at a similar location. Hence the winter event might have been unprecedented by yet another measure, as polynya events have never repeated within a year over the study region except during 2018.

## 6 Summary

Following the previous studies of the 2018 winter polynya by Moore et al. (2018) and Ludwig et al.
(2019), this study additionally demonstrates the cause and evolution of the three observed winter open

water events that ever occurred under a specific wind pattern, which removes sea ice mechanically out of the region: i.e., December 1986 as well as February 2011 and 2017. In all of the winter polynya events, the wind direction was primarily from the south and southwest (Figs. 4, 6, and 9). The size of the polynya depended on the strength and persistence of southerly winds, and the polynya closure was a result of the relaxation of these conditions. Although the atmospheric condition was associated with SSW in February 2018, other three winter polynyas occurred in non-SSW winters. The polynya was larger in February 2011 compared to the one in 2017 because wind duration was longer: four days of strong southerly wind ($> 10$ m s$^{-1}$) in 2011 compared to two days in 2017 (Fig. 8a). Hence, 31 % more sea ice was removed in the former, although wind speed and wind stress were similar between the two years (Table 1). The observed polynya in February 2018 was the largest one over the satellite SIC record period since its inception in 1979. This resulted from much stronger and more persistent southerly-southeasterly winds (Fig. 4g) off the coast of northern Greenland than the previous events. Although the southerly wind in December 1986 was as strong as in February 2018, its duration was shorter, and thus not only the size of the polynya, but also the turbulent heat flux was relatively smaller. Sea ice was significantly thicker in the 1980s north of Greenland, but polynyas were not necessarily prevalent with thinner ice in the winters of 2015–2025 from the RASM-DPLE ensembles. Out of 100 winters, 17 and 16 winter polynyas were produced in the two RASM-DPLE ensembles initialized 30 years apart in December 1985 and 2015, respectively. Given the rate of winter polynya occurrence and its apparent lack of sensitivity to the initial sea ice thickness, we conclude that a dominant cause of these winter polynyas stems from internal variability of atmospheric forcing rather than from the forced response to a warming climate.

**Code/Data availability**

All RASM simulations are archived in the HPCMP archive system and will be available upon publication according to the U.S. DoD data policy.

**Author contributions**

Y. L. and W. M. conceived the study and wrote the paper. A. C., S. K., R. O., and M. S. undertook the implementation and development of the RASM simulation. J. C., J. C. K., J. S., and H. W. contributed to the interpretation of results. All authors except S. K. and A. C. commented on the manuscript.

**Competing interests**

The authors declare no competing interests.

## Acknowledgements

This work was funded by NSF OPP IAA1603602 (NPS), DOE RGMA DE-SC0014117 (NPS), NSF OPP-1603544 (CU Boulder), the Canada 150 Chair program (J. S.), and the Ministry of Science and Higher Education in Poland (R. O.) in the frame of international project agreement 3808/FAO/2017/0. The PNNL is operated for the U.S. DOE by Battelle Memorial Institute under contract DE-AC05-76RL01830. In addition, computing resources were provided by the U.S. DOD High Performance Computer Modernization Program (HPCMP). The merging of CryoSat-2 and SMOS data was funded by the ESA project SMOS & CryoSat-2 Sea Ice Data Product Processing and Dissemination Service and data from February 2011 to 2018 were obtained from https://www.meereisportal.de (grant: REKLIM-2013-04). We acknowledge the use of imagery from the NASA Worldview application (https://worldview.earthdata.nasa.gov), part of the NASA Earth Observing System Data and Information System (EOSDIS). Finally, we thank the reviewers for providing helpful comments, which improved an earlier version of this publication.

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

**Table 1. Characteristics of the polynya events, when daily dynamic sea ice removal is greater than 10 km$^3$ d$^{-1}$ (i.e., dynamic volume tendency (DVT) ≤ –10 km$^3$ d$^{-1}$) for more than three consecutive days, and the corresponding wind condition near a weather station, as well as sea ice thickness (SIT) over northern Greenland from the RASM hindcast simulation (1980–2020).**

| Year | Date when DVT ≤ –10 km$^3$ d$^{-1}$ (No. of polynya days) | Five-day mean SIT (m) before polynya | Daily Mean DVT (km$^3$ d$^{-1}$) | Total sea ice removal (km$^3$) | Relative total sea ice removal (%)[*] | North-South wind during polynya | | | Nearest weather station |
|------|------|------|------|------|------|------|------|------|------|
| | | | | | | Time-integrated Pseudo–wind stress[**] (m$^2$ s$^{-2}$ d) | Mean speed (m s$^{-1}$) | Max. speed (m s$^{-1}$) | |
| 1986 | 12-16 Dec (5 days) | 4.4 | –28.4 | –142 | 27.5 | 717 | 11.0 | 17.6 | Cape Morris Jessup |
| 2011 | 12-15 Feb (4 days) | 3.6 | –13.8 | –55 | 13.0 | 476 | 10.2 | 15.1 | Station Nord |
| 2017 | 8-10 Feb (3 days) | 3.3 | –14.0 | –42 | 10.8 | 354 | 9.5 | 15.3 | Station Nord |
| 2018 | 16–25 Feb (10 days) | 3.2 | –18.9 | –189 | 50.3 | 1486 | 11.3 | 21.8 | Cape Morris Jessup |

[*]= total sea ice removal ÷ (five-day mean SIT × study area) where the study area is 117,390 km$^2$
[**]only for the northward wind component

**Table 2. Daily mean net turbulent (sensible and latent) heat flux (W m$^{-2}$) and its standard deviation (s.d.) during the polynya events over open water areas (km$^2$), where sea ice thickness (SIT) is less than 10 cm, from the RASM hindcast and DPLE simulations. The negative values indicate heat loss from the polynya to the atmosphere.**

| RASM Cases | Year | Date | Daily Turbulent Flux (W m$^{-2}$) | | | Daily Open Water Area (km$^2$) | | | Total integrated turbulent heat (W) |
|---|---|---|---|---|---|---|---|---|---|
| | | | max | mean | s.d. | max | mean | s.d. | |
| Hindcast | 1986 | 14—18 Dec. | −48.8 | −18.9 | 18.5 | 7,811 | 4,120 | 2,822 | −3.56 × 10$^{11}$ |
| | 2011 | 15—18 Feb. | −116 | −91.6 | 21.7 | 601 | 322 | 236 | −1.25 × 10$^{11}$ |
| | 2017 | 10 Feb. | −1.59 | −1.59 | n/a | 172 | 172 | n/a | −2.74 × 10$^{8}$ |
| | 2018 | 20 Feb.—1 Mar. | −124 | −60.9 | 39.4 | 13,000 | 4,210 | 4,390 | −2.03 × 10$^{12}$ |
| D P L E — 1985-initialized ensemble #4 | 1988 | 17—21 Jan. | −194 | −103 | 79.8 | 429 | 309 | 130 | −1.54 × 10$^{11}$ |
| 2015-initialized ensemble #2 | 2024 | 20—24 Jan. | −110 | −74.1 | 23.8 | 6,090 | 2,540 | 2,340 | −8.87 × 10$^{11}$ |

**Table 3.** The potential polynya events defined when daily dynamic sea ice removal is greater than 10 km³ d⁻¹ (i.e., dynamic volume tendency (DVT) ≤ −10 km³ d⁻¹) for more than three consecutive days, and the corresponding wind condition near Cape Morris Jessup as well as sea ice thickness (SIT) over northern Greenland from the RASM-DPLE simulation, initialized on 1 December 1985. A correlation coefficient ($r$) is calculated between daily DVT and North-South wind for December–March and the bold indicates the largest polynya event.

| DPLE ensemble member | Year | Month | No. of days when DVT ≤ −10 km³ d⁻¹ | Five-day mean SIT (m) before polynya | Daily mean DVT (km³ d⁻¹) | Total sea ice removal (km³) | Relative total sea ice removal (%)* | North-South wind during polynya | | | $r$ ($p<0.01$) between daily DVT and North-South wind (Dec−Mar) |
| | | | | | | | | Time-integrated pseudo–wind stress** (m² s⁻² d) | Mean speed (m) | Max. speed (m) | |
|---|---|---|---|---|---|---|---|---|---|---|---|
| 1 | 1986 | Jan | 3 | 4.1 | −16.1 | −48 | 10.0 | 307 | 9.7 | 13.6 | −0.73 |
| 1 | 1986 | Mar | 4 | 4.4 | −12.6 | −50 | 9.7 | 31 | 2.5 | 4.3 | −0.73 |
| 1 | 1988 | Mar | 3 | 4.0 | −11.5 | −35 | 7.5 | 168 | 7.4 | 9.9 | −0.76 |
| 1 | 1990 | Mar | 4 | 3.5 | −15.6 | −62 | 15.1 | 395 | 8.6 | 15.5 | −0.72 |
| 1 | 1991 | Dec | 3 | 3.8 | −17.6 | −53 | 11.9 | 244 | 8.6 | 12.2 | −0.66 |
| 2 | 1991 | Dec | 6 | 3.8 | −22.3 | −134 | 30.0 | 761 | 10.6 | 14.7 | −0.81 |
| 3 | 1987 | Jan | 4 | 3.8 | −21.4 | −85 | 19.1 | 475 | 10.1 | 19.0 | −0.69 |
| 4 | 1986 | Dec | 6 | 3.7 | −16.9 | −101 | 23.3 | 439 | 7.6 | 15.2 | −0.78 |
| **4** | **1988** | **Jan** | **7** | **4.2** | **−17.2** | **−120** | **24.3** | **438** | **7.6** | **12.0** | **−0.64** |
| 4 | 1989 | Feb | 5 | 4.0 | −20.7 | −104 | 22.1 | 651 | 11.2 | 15.7 | −0.70 |
| 4 | 1991 | Feb | 3 | 3.2 | −25.2 | −76 | 20.2 | 550 | 12.6 | 17.8 | −0.81 |
| 4 | 1991 | Dec | 3 | 3.6 | −13.5 | −40 | 9.5 | 230 | 8.3 | 12.9 | −0.78 |
| 5 | 1986 | Jan | 4 | 3.8 | −17.0 | −68 | 15.2 | 260 | 7.3 | 13.7 | −0.70 |
| 5 | 1987 | Jan | 4 | 3.5 | −16.5 | −66 | 16.1 | 121 | 4.9 | 8.5 | −0.68 |
| 5 | 1987 | Dec | 4 | 3.4 | −13.9 | −55 | 13.8 | 438 | 10.1 | 14.1 | −0.61 |
| 5 | 1988 | Mar | 3 | 3.8 | −20.0 | −60 | 13.5 | 121 | 4.8 | 11.2 | −0.61 |
| 6 | 1986 | Dec | 3 | 3.4 | −12.6 | −38 | 9.5 | 111 | 5.6 | 10.0 | −0.74 |
| 6 | 1994 | Feb | 4 | 2.9 | −22.0 | −88 | 25.8 | 700 | 13.0 | 16.7 | −0.73 |
| 7 | 1990 | Dec | 3 | 3.3 | −20.8 | −62 | 16.0 | 283 | 9.4 | 12.6 | −0.73 |
| 7 | 1994 | Jan | 3 | 3.8 | −14.7 | −44 | 9.9 | 255 | 8.5 | 12.3 | −0.65 |
| 8 | 1992 | Jan | 3 | 4.3 | −13.6 | −41 | 8.1 | 277 | 9.2 | 15.0 | −0.71 |
| 8 | 1995 | Jan | 3 | 3.3 | −16.9 | −51 | 13.2 | 172 | 6.9 | 13.0 | −0.63 |
| 9 | 1989 | Dec | 4 | 4.0 | −13.2 | −53 | 11.3 | 228 | 7.1 | 10.9 | −0.71 |
| 9 | 1995 | Mar | 3 | 3.8 | −16.3 | −49 | 11.0 | 322 | 10.2 | 11.6 | −0.66 |
| 10 | 1987 | Feb | 3 | 3.6 | −13.2 | −40 | 9.5 | 145 | 6.6 | 10.1 | −0.62 |
| | | | mean | 3.7 | - | 65 | - | 325 | 8.3 | - | - |
| | | | standard deviation | 0.36 | - | 27 | - | 196 | 2.4 | - | - |

*= total sea ice removal ÷ (five-day mean SIT × study area) where the study area is 117,390 km²

** only for the northward wind component

**Table 4. Same as Table 3, but for the RASM-DPLE simulation, initialized on 1 December 2015.**

| DPLE ensemble member | Year | Month | No. of days when DVT ≤ −10 km³ d⁻¹ | Five-day mean SIT (m) before polynya | Daily mean DVT (km³ d⁻¹) | Total sea ice removal (km³) | Relative total sea ice removal (%)* | North-South wind during polynya | | | r (p<0.01) between daily DVT and North-South wind (Dec−Mar) |
|---|---|---|---|---|---|---|---|---|---|---|---|
| | | | | | | | | Time-integrated pseudo–wind stress** (m² s⁻² d) | Mean speed (m) | Max. speed (m) | |
| 1 | 2016 | Feb | 5 | 2.7 | −14.4 | −72 | 22.7 | 165 | 5.0 | 10.1 | −0.75 |
| 1 | 2022 | Feb | 3 | 2.6 | −15.1 | −45 | 14.7 | 216 | 8.2 | 11.5 | −0.62 |
| 1 | 2023 | Jan | 3 | 2.5 | −14.4 | −43 | 14.7 | 321 | 10.0 | 14.3 | −0.78 |
| 1 | 2023 | Jan | 3 | 1.8 | −15.4 | −46 | 21.8 | 299 | 9.8 | 13.1 | −0.78 |
| 2 | 2016 | Jan | 3 | 2.8 | −16.4 | −49 | 14.9 | 192 | 7.8 | 10.9 | −0.76 |
| **2** | **2024** | **Jan** | **7** | **3.3** | **−19.4** | **−136** | **35.1** | **841** | **9.8** | **19.2** | **−0.78** |
| 4 | 2019 | Mar | 3 | 3.0 | −15.4 | −46 | 13.1 | 235 | 7.4 | 16.9 | −0.63 |
| 5 | 2018 | Jan | 5 | 3.0 | −20.8 | −104 | 29.5 | 740 | 10.8 | 17.9 | −0.77 |
| 5 | 2018 | Mar | 3 | 3.0 | −15.4 | −46 | 13.1 | 129 | 5.6 | 12.9 | −0.77 |
| 5 | 2020 | Feb | 5 | 3.0 | −15.5 | −77 | 21.9 | 379 | 8.4 | 12.2 | −0.75 |
| 6 | 2016 | Jan | 4 | 2.8 | −12.1 | −48 | 14.6 | 379 | 9.6 | 12.3 | −0.77 |
| 6 | 2018 | Dec | 3 | 3.0 | −15.1 | −45 | 12.8 | 358 | 10.4 | 16.2 | −0.71 |
| 6 | 2024 | Jan | 4 | 2.8 | −20.0 | −80 | 24.3 | 352 | 8.1 | 15.2 | −0.72 |
| 7 | 2023 | Dec | 4 | 3.4 | −16.4 | −66 | 16.5 | 305 | 8.2 | 14.8 | −0.70 |
| 8 | 2017 | Feb | 3 | 2.9 | −14.5 | −43 | 12.6 | 558 | 13.1 | 17.0 | −0.79 |
| 8 | 2025 | Jan | 5 | 2.8 | −18.5 | −92 | 28.0 | 531 | 9.9 | 15.8 | −0.76 |
| | | | mean | 2.8 | - | 65 | - | 375 | 8.9 | - | - |
| | | | standard deviation | 0.37 | - | 28 | - | 201 | 2.0 | - | - |

*= total sea ice removal ÷ (five-day mean SIT × study area) where the study area is 117,390 km²
** only for the northward wind component

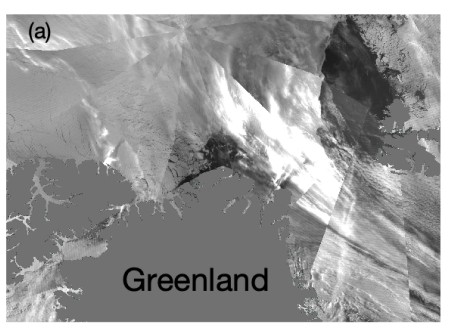

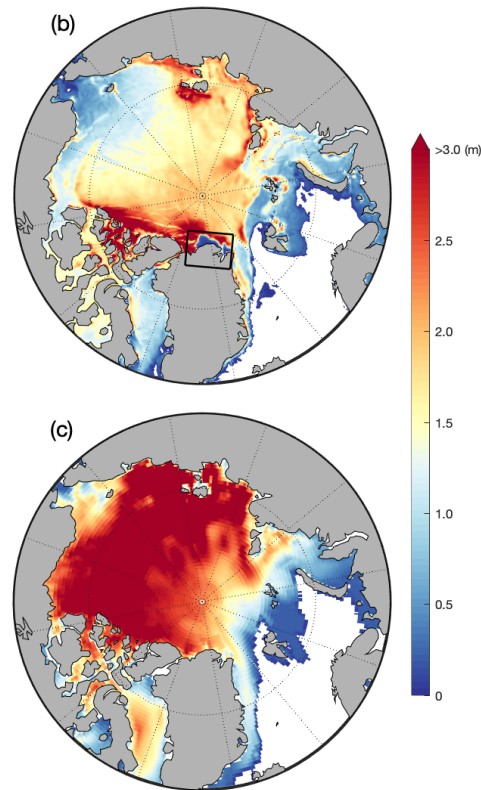

Figure 1. (a) The Visible Infrared Imaging Radiometer Suite (VIIRS) Nighttime Imagery (Day/Night Band, Enhanced Near Constant Contrast) over northern Greenland on 25 February 2018 (adapted from NASA Worldview at https://worldview.earthdata.nasa.gov) and sea ice thickness from (b) the RASM hindcast simulation as well as (c) CFS version 2 reanalysis over the Arctic on 25 February 2018. The rectangular box indicates the polynya area.


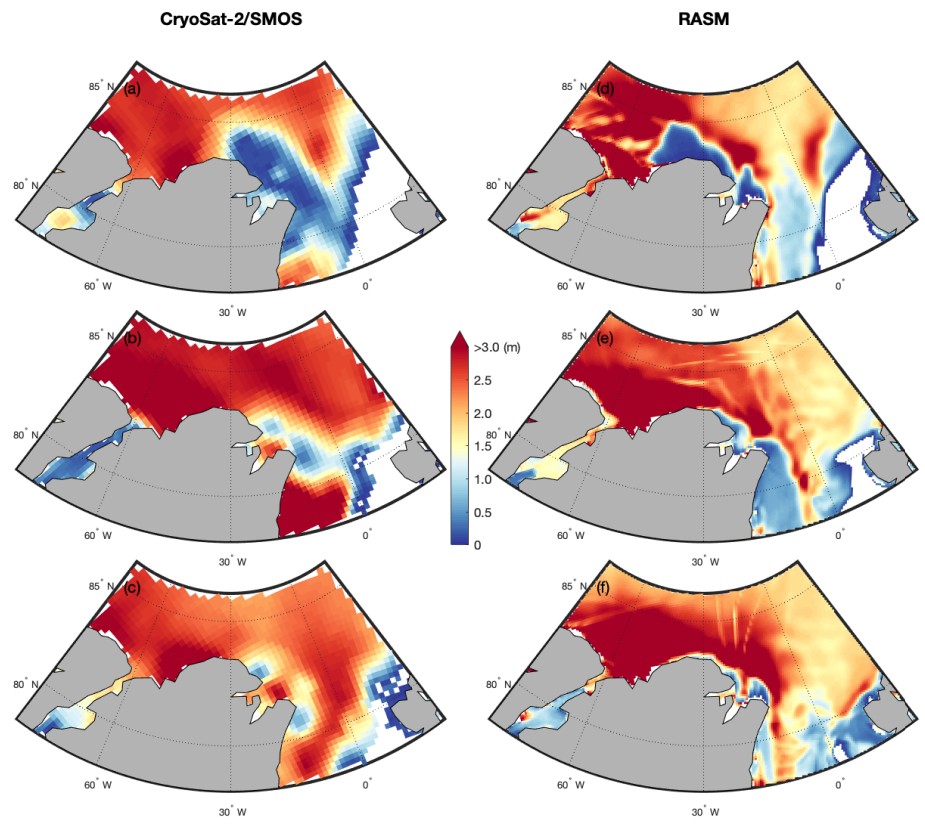

**Figure 2. Mean sea ice thickness (SIT; m) over the northern Greenland region from the CryoSat-2/SMOS merged product on (a) 22–28 February 2018, (b) 8–14 February 2017, and (c) 13–19 February 2011 as well as daily SIT from the RASM hindcast simulation on (d) 25 February 2018, (e) 11 February 2017, and (f) 16 February 2011.**

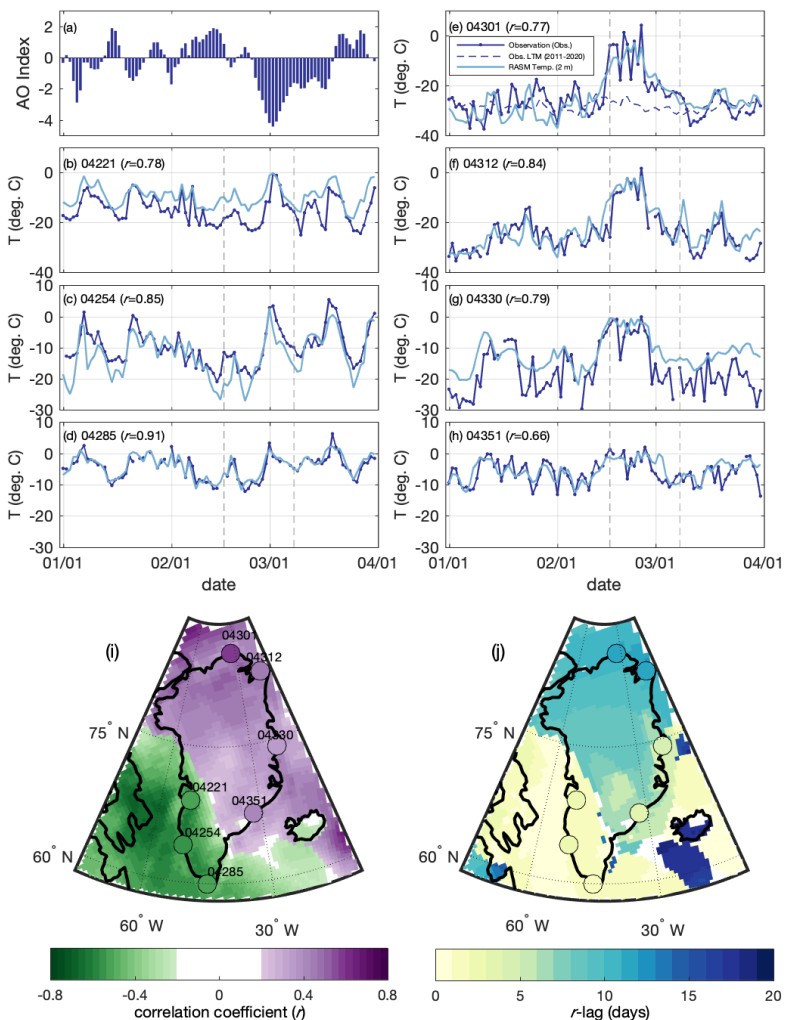


**Figure 3. (a)** The observed daily AO index and **(b)-(h)** the daily mean near-surface air temperature (T; ⁰C) (blue) during January–March 2018 from the selected weather stations around Greenland as shown in (i) and (j) as circle markers. The RASM daily 2 m air temperature (light blue) is from the nearest grid cell and the observed long-term (2011–2020) daily mean (LTM) (blue-dashed)

is only at Station 04301. The correlation coefficient (r) is statistically significant (p<0.05) between the observed and the simulated air temperature is shown in the upper left. The vertical dashed lines indicate the start and end dates of the 2018 winter polynya period. The spatial maps show (i) the maximum correlation coefficients and (j) their lagged timescales (days) between the AO index and the RASM air temperature. Note that grid cells not statically significant (p>0.05) are indicated as white. The color scale of the markers quantifies the relationship between the AO index and the observed air temperature at each weather station.

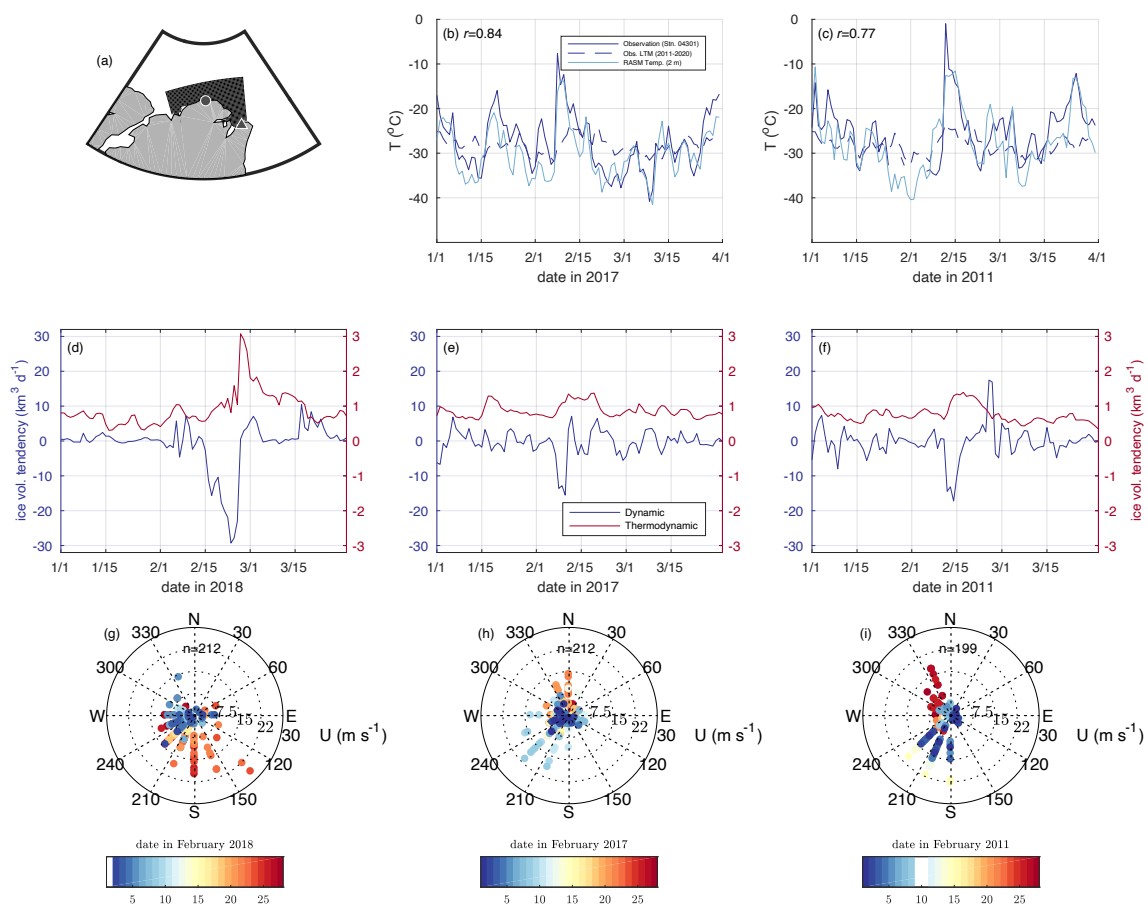


**Figure 4.** The observed daily near-surface air temperature (T; °C) (blue) from Station 04301 (Cape Morris Jesup as ●) in (a) and the RASM daily 2 m air temperature (light blue) from the nearest grid cell during January−March of (b) 2017 and (c) 2011; the observed long-term (2011-2020) daily mean (LTM) (blue-dashed). The correlation coefficient (*r*) between the observed and the simulated air temperature is shown in the upper left. The time rate of change of the RASM daily sea ice volume (km³ d⁻¹) due to thermodynamic (red) and dynamic (blue) tendency during January−March of (d) 2018, (e) 2017, and (f) 2011, spatially integrated for the black-shaded area in (a). The wind rose plots of 3-hourly wind data at Station 04312 (Station Nord as ▲) during February of (g) 2018, (h) 2017, and (i) 2011: wind speed (U; m s⁻¹) and direction. The missing data are indicated as white in each color bar.

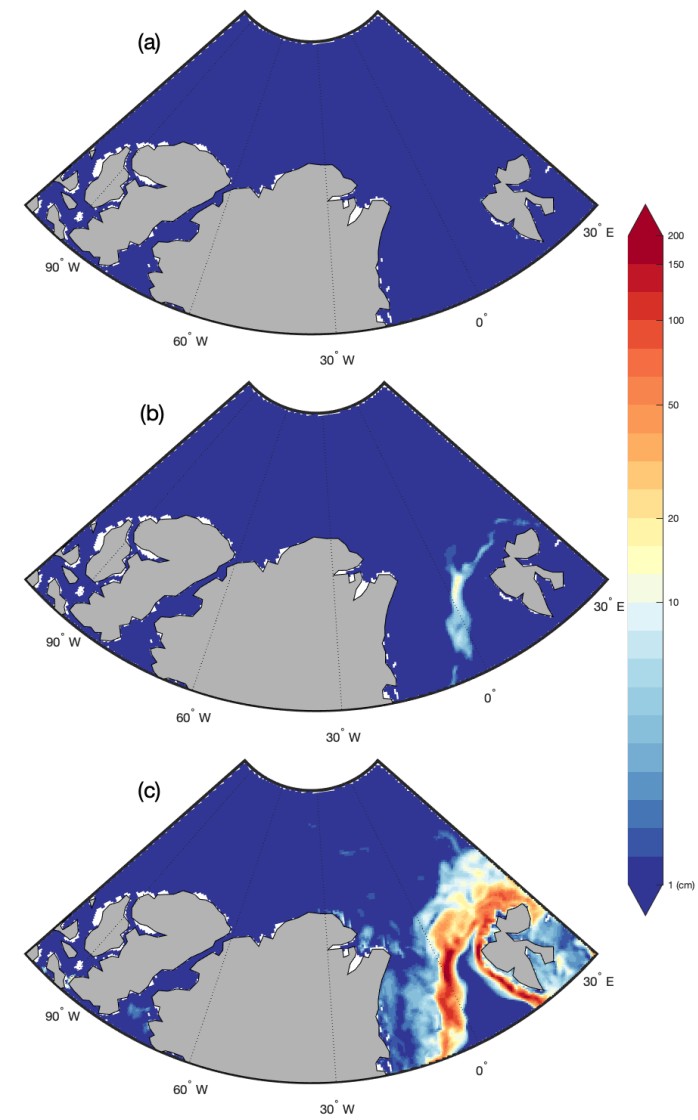


**Figure 5. Monthly integrated sea ice melt terms over the northern Greenland region during February 2018 from the RASM hindcast simulation: (a) surface top, (b) lateral, and (c) basal melt (cm).**

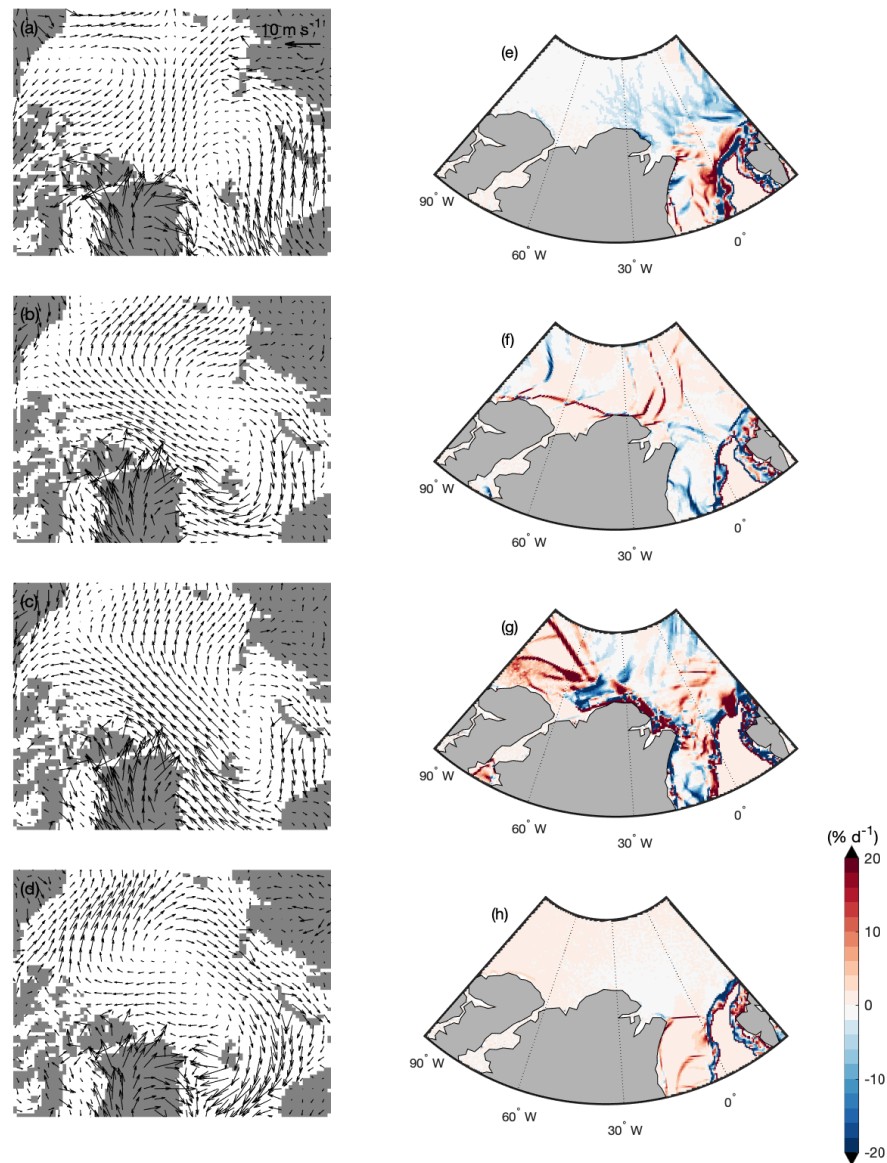

 **Figure 6. Four major patterns with self-organizing maps (frequency of occurrence as %) of the near-surface 6-hourly wind fields from the RASM simulation during the time periods including before and after the 2018 polynya event: (a) 5–11 February (93 %), (b) 15–20 February (83 %), (c) 20–26 February (93 %), and (d) 3–13 March (95 %). The wind vectors were sub-sampled at every three grid cells for a plotting purpose. RASM daily sea ice divergence (% d$^{-1}$) within each period above is shown on (e) 10 February, (f) 16 February, (g) 24 February, and (h) 4 March 2018.**

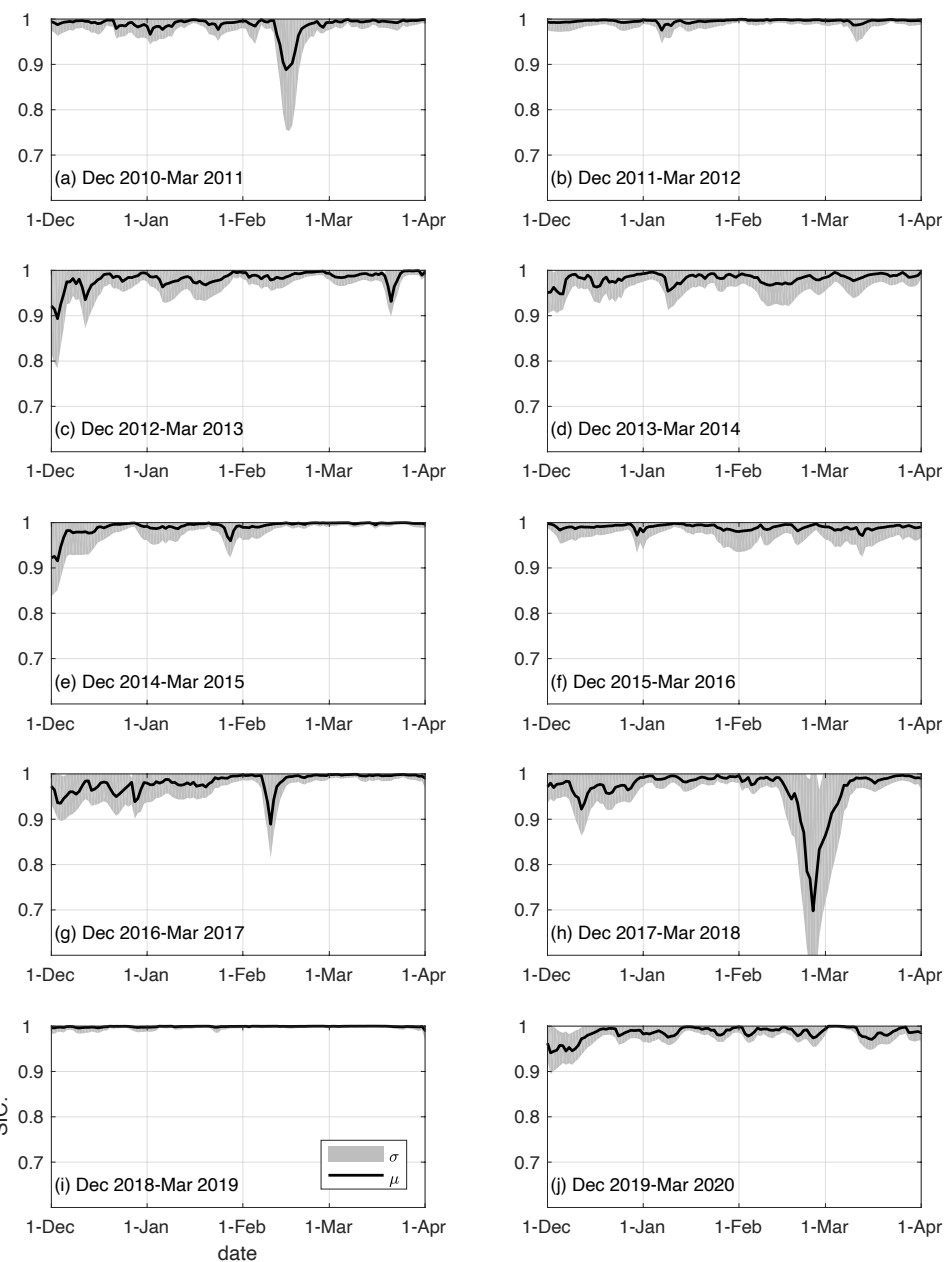


**Figure 7. (a)-(j) Satellite-derived (NASA team algorithm) daily mean (µ) sea ice concentration (SIC) (black) for the northern Greenland region (see Fig. 4a) during January−March from 2011 to 2020. The grey shading depicts one standard deviation (σ) (gray) from the mean SIC.**

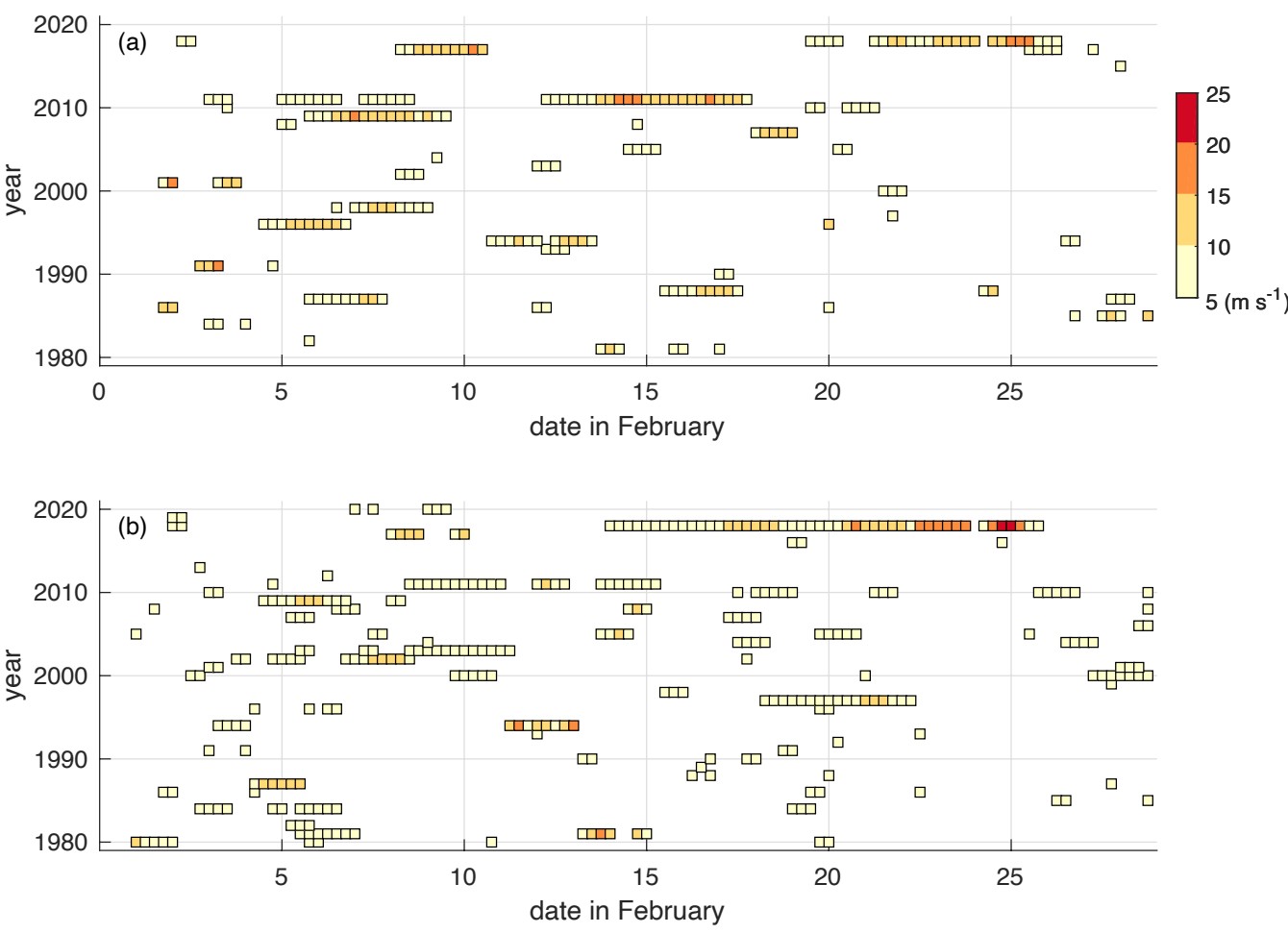


**Figure 8. The northward wind component greater than 5 m s⁻¹ from the RASM hindcast simulation during February of 1980−2020 at the nearest grid to (a) Station 04312 (Station Nord) and (b) Station 04301 (Cape Morris Jesup). Each square represents the 6-hourly surface wind, and its color indicates the magnitude of wind speed.**


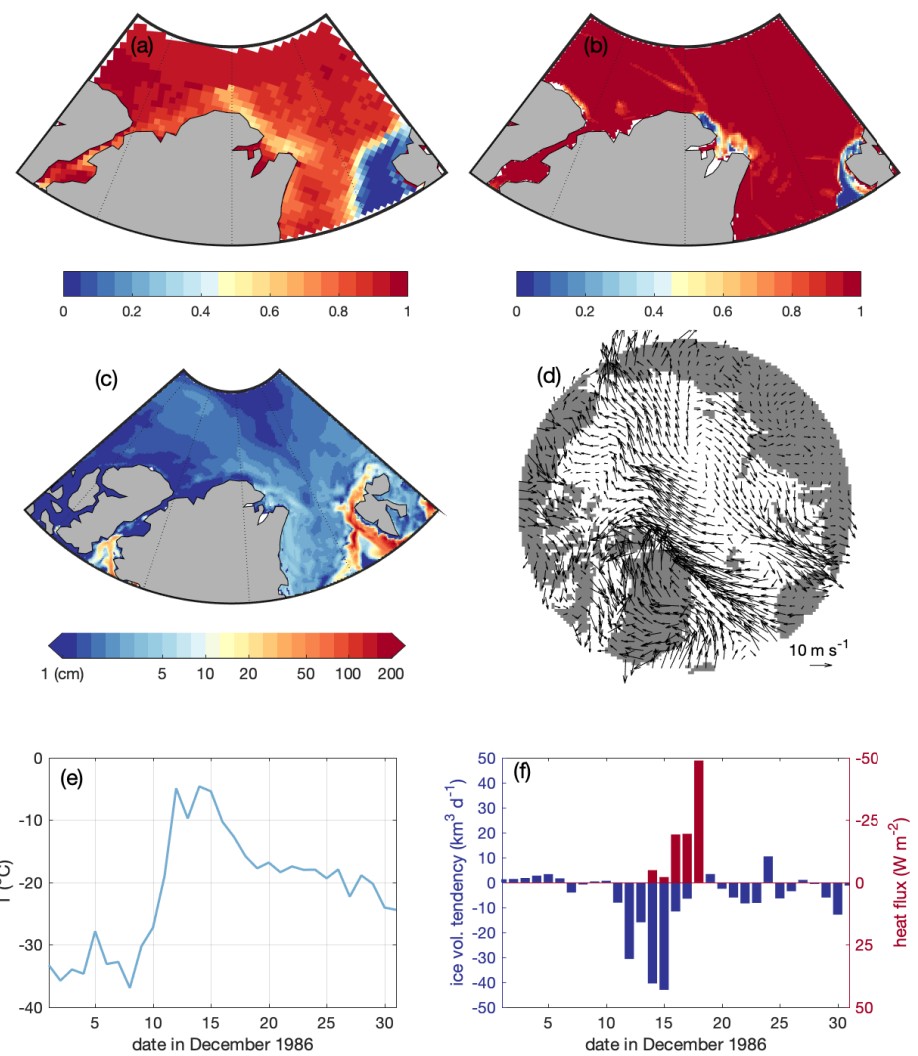

**Figure 9. Mean sea ice concentration (NASA Team algorithm) on 15 December 1986 over the northern Greenland region from (a) the Nimbus-7 SMMR and (b) the RASM hindcast simulation. (c) Monthly integrated sea ice melt (top, lateral, and basal) for December 1986 and (d) near-surface (at 10 m) wind fields on 12 December 1986 from the RASM hindcast simulation. (e) RASM daily near-surface air temperature (T; ºC) from the nearest grid cell to Station 04301 (Cape Morris Jesup) (f) Time rate of change of the RASM daily sea ice volume (km³ d⁻¹) (blue) due to dynamic tendency during December 1986, spatially integrated for the region shown in Fig. 4a. Daily mean net turbulent (sensible and latent) heat flux (W m⁻²) (red) is also shown over the same region where sea ice thickness is less than 10 cm from the RASM hindcast simulations. The negative values indicate loss from the region.**


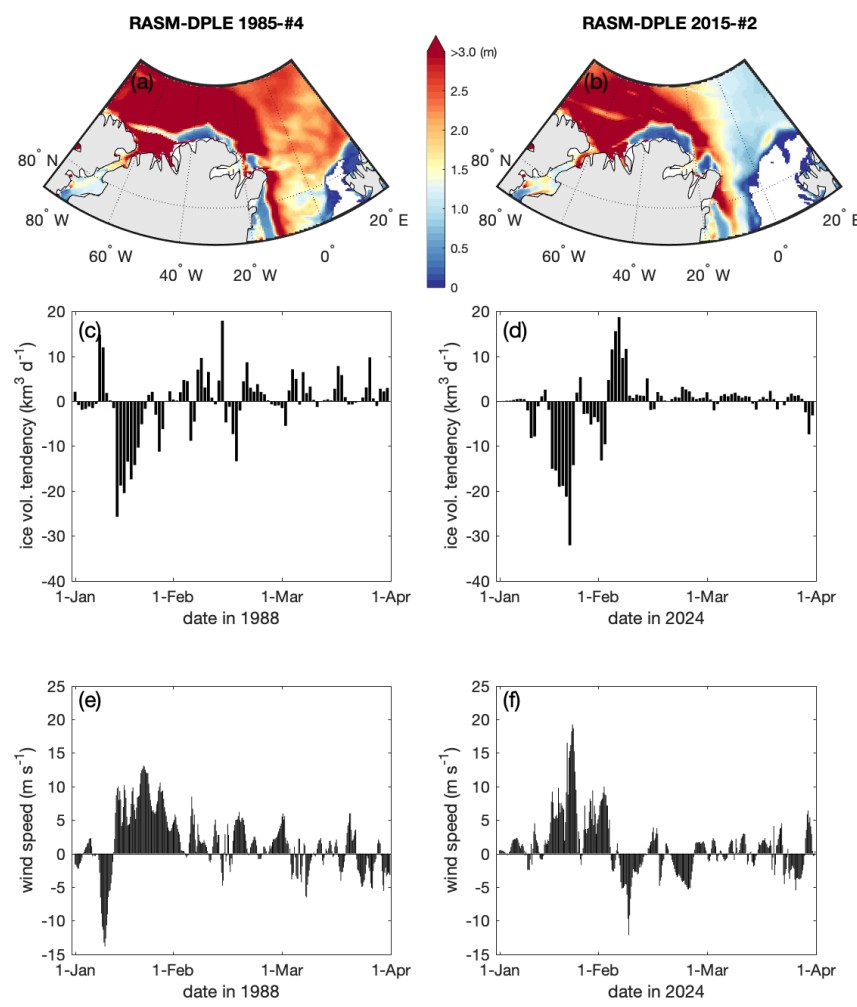

**Figure 10. Sea ice thickness (SIT; m), sea ice dynamic volume tendency (DVT; km³ d⁻¹) for the northern Greenland region (see Fig. 4a), and north(positive)-south(negative) wind component (m s⁻¹) during winter months from the ensemble member #4 in the 1985-initialized run and the ensemble member #2 in the 2015-initialized run: SIT on (a) 22 January 1988 and (b) 14 January 2024, daily**
**DVT in January–March of (c) 1988 and (d) 2024, and six-hourly north-south wind in January–March of (e) 1988 and (f) 2024.**