# Peer review of "Causes and Evolution of Winter Polynyas North of Greenland"

_The Cryosphere, 2021_

## Referee Comment (RC1)

Review on "Causes and Evolution of Winter Polynyas over North of Greenland" by Younjoo J. Lee, Wieslaw Maslowski, John J. Cassano, Jaclyn Clement Kinney, Anthony P. Craig, Samy Kamal, Robert Osinski, Mark W. Seefeldt, Julienne Stroeve, Hailong Wang.

The paper describes the performance of the fully-coupled Regional Arctic System Model (RASM) with respect to the simulation of polynya events north of Greenland. A 42-year long simulation (1979-2020) is analysed in combination with satellite products and weather station data. Additionally, two ensembles are generated by forcing RASM with output from the Community Earth System Model (CESM) Decadal Prediction Large Ensemble (DPLE) simulations. The two ensembles, initialized in December 1985 and December 2015, are investigated with respect to precondition of winter polynya events.

The paper describes a nice application of dynamical downscaling. However the paper needs major revision mainly because of two main points of criticism:

(1) It is not immediately clear what the added value of this paper in comparison to the papers of Moore et al. (2018) and Ludwig et al. (2029) is. In both of the latter papers sea ice-ocean models (PIOMAS in case of Moore et al. and NAOSIM in case of Ludwig et al.) are used to analyse the polynya event in more detail as possible with observations alone with almost identical findings (e.g. that preconditioning has no effect on the polynya event in 2018). I suggest to revise the manuscript carefully to make clearer the scientific added value of this study.

(2) Unfortunately, winter is defined in this study from January to March excluding December. If December would have been included the authors would have not missed the polynya north of Greenland in December 1986 – Moore et al. missed it as well because they concentrated on February only (see plot below based on own unpublished analyses). The plot shows the ice concentration as modelled by NAOSIM (left panel) and as observed by satellite observations (OSI SAF – right panel) on 15[th] of December 1986

ice concenztation 1986m12d15 'f'        OSI SAF (%) 19861215

Inspection of the wind field in December 1986 north of Greenland in reveals a northward wind anomaly of almost the same strength and duration as in February 2018 (plot below)

[Figure]

However, there is a dramatic difference. While the occurrence of the 2018 polynia coincides with a sudden stratospheric warming (SSW) event a few days earlier (and associated with a strong decrease in the NAO) the 1986 event does not show any SSW nor any strong NAO change (plot below – left: u-zonal 10mb in February 2018 (top) and in December 1986 (bottom); right: NAO) (based on NCEP/CSFR/CSFv2).

[Figure]

RASM might be an ideal tool to analyse this event in 1986 as well with respect to the processes in the atmosphere. (The reason why the own research presented above was never published is that we had no appropriate fully-coupled model at hand to perform a thorough analysis in the atmosphere (and probably to perform some sensitivity studies with such a model)).

Beside this two major points of criticism I listed below a number of points the authors are ask to take into account in the revision. The importance of my suggestions is indicated by minor/major in front of each item but follows the order in the manuscript.

1. Minor - line 49 'Introduction': Some of the citations given are pretty old and should be replaced by newer publications. One could be https://journals.ametsoc.org/view/journals/clim/34/13/JCLI-D-20-0848.1.xml

2. Major – line 60: Figure 1 needs some heavy revision. Panel a) is too dark. Panel b) and c) should be shown in a similar projection as a) to make the comparison easier. The rectangle in b) and c) does not compare well will panel a). Obviously RASM is not able to reproduce the large area of open water north of Fram strait that can be seen in the observations (cmp. plot below). This should be discussed in the text and reasons for the deficit should be given (certainly shortcomings in the vertical mixing of the ocean model). The SIT from CFRv2 in panel c) is very unrealistic. A brief discussion on the reliability of SIT from CFRv2 is necessary if the plot should be shown.

[Figure]

3. Minor - line 63 'Introduction': I was very surprised to see no hint to the SSW when Moore et al. is cited. The coincidence of the polynia with the SSW is mentioned later under 'Discussion' but I would prefer to have some statements about the possible connection to the SSW when citing Moore et al. for the first time because this is the strongest message in

that paper.

4. Minor but important  - line 124: CS2SMOS should be referenced correctly. From meereisportal.de:

> For all CryoSat-2/SMOS data, please
>
> 1) include the following phrase into the acknowledgment:
>
> The merging of CryoSat-2 und SMOS data was funded by the ESA project SMOS & CryoSat-2 Sea Ice Data Product Processing and Dissemination Service and data from DATE to DATE were obtained from https://www.meereisportal.de (grant: REKLIM-2013-04).
>
> 2) refer to: Ricker, R.; Hendricks, S.; Kaleschke, L.; Tian-Kunze, X.; King, J. and Haas, C. (2017), A weekly Arctic sea-ice thickness data record from merged CryoSat-2 and SMOS satellite data, The Cryosphere, 11, 1607-1623, doi:10.5194/tc-11-1607-2017.

5. Minor – line 177: Fig. 3 is referenced before Fig. 2. Check the order of the plots. It makes the manuscript unnecessary complicated to read.

6. Major – line 179: "... to early March **is** captured well ...". I disagree with the statement (see comment below – line 216).

7. Major – line 216:  "The RASM's realistic representation of the polynya ...". What seems to be realistic is the size of the polynya but not the location which is very disappointing for a downscaling system. Fig. 2 shows very convincing that the polynya is located too far to the west – the largest fraction of the polynya is located in areas where CS2SMOS shows thicknesses of more than about 1m! Reasons for the mislocation of the polynya should be discussed. In Ludwig et al. a sound estimate of the size of the polynya is given (about 600.000 km2 in maximum) but the size of the polynia in RASM is not compared to this number. This should be done! Instead modelled volume growth rates are discussed for which no observation analog exists (CS2SMOS based estimates are definitely to uncertain given the very coarse resolution). In Ludwig et al. thermodynamical growth rates based on simple estimates are given which should be discussed together with the estimates from RASM.

8. Minor – line 232: "... removal due to the polynya ..." Not the polynya is removing the sea ice but the winds are removing the sea ice and forming the polynia. Please revise.

9. Minor – line 237: I do not understand what should be learned from whole subsection.

10. Major – line 260: The whole subsection 4 might need a revision in the light of the strong wind event in December 1986 mentioned above.

11. Major – line 367 'Discussion': Obviously the whole subsection needs reformulation after inspection of the December 1986 wind/polynya event.

---

## Referee Comment (RC2)

The paper studies winter polynyas in North Greenland.  A modeling study using a fully coupled downscaling framework examines the roles of internal variability and long-term sea ice thinning. An examination of the causes and consequences of polynya development on aspects on the energy balance is presented. Model hindcasts are also compared to observations for validation.  The key results are that a) the model reproduces key features of three major winter polynyas in the decade of 2010-2020 as well as their meteorological drivers (northward winds>latent heat polynya), b) long term thinning of sea ice did not significantly contribute to their creation, while c) an increase in frequency of wind events that are capable of creating winter polynyas north of Greenland may be responsible for the fact that major events have all occurred within the last decade.

While some of the results are not new, they replicate previous findings (Moore et al. 2018, Ludwig et al. 2019). In particular they use a coupled atmosphere-ice-ocean modelling framework that overcomes some of the limitations of the ice-ocean framework used in Moore et al. 2018 for testing the impact of sea ice thinning. The analysis regarding the increasing frequency of Polynya maker winds seems also new.  I think the paper is generally well conceived and executed and makes a significant contribution to the field and I have no requests for major changes.  The paper might perhaps benefit from some reorganization that better separates the strongly supported results (lack of role of thinning) from the more "future research is needed" ones (increase in polynya frequency due to wind changes). But that's a matter of taste (and likely personal bias) and I think the paper has lingered in reviewer space for too long already so I don't think that should be a requirement.

Details:

Line 80: Figure 1c. The CFS sea ice thickness looks terrible. Since you are using only above 540 mb probably not an issue for you, but maybe worth a note.

Line 123… "the mean SIC was used to detect … when it dropped below 90%"
I think this needs to be justified since the selection of this threshold probably affects the number of polynyas you would have detected? Does the 90% reflect some kind statistical threshold of variability or a value in the literature?  This definition probably has to remain somewhat arbitrary but the sensitivity of results to this selection should be discussed somewhere in the paper.

Line 165 ..SOMs. Seems to me that the SOM analysis is a bit of an overkill in this context and may add more confusion than explanation. I think simple wind (anomaly) composites for the Polynya events would have done the job. I know, you did the work and hey, ML! but maybe save for SOMe other paper?

Line 240 . Looks like Figure 6 is mentioned before Figure 5 (and I don't see a reference to Figure 5). Figure 5 isn't very interesting at this scale anyhow.

Figure 6. Please increase the size. This is hard to see for my aging eyes unless this is improved in production.  Also, do we really need to see the full Arctic pattern or is a smaller cut out region sufficient to see what is relevant. What is the significance of the result that this temporal evolution goes through a number of SOM patterns? The key part is northward winds in the region, isn't it?

Fig 7 (S3,S4). Couldn't those be condensed a bit to highlight the years that should be highlighted? What is the grey shading for the standard deviation? I would have expected the standard deviation of the SIC

for that day in the full time series. Why is this larger for the "polynya" cases? See also above comment on defining the SIC threshold. I would have thought a reasonable definition would be 1 or 2 sigma of interannual variability (or quartiles or something like that)?

Line 275: Polynya periods… defined as .

How does this interact with your previous definition of 90% SIC. Needs some clarification.

Line 290.. This section could perhaps get a separate heading "What's driving changes in polynya Frequency".

Totally a style thing, but I would lead with testing the hypothesis that is rejected (thinning ice) and corroborates previous results (Moore et al. 2018) and follow with what the more likely hypotheses of "changes in winds". That would also allow a natural transition to a discussion of the "unanswered" question left for further research (why is the wind changing?).

I like Fig 8 but think that if the increased frequency in wind events is considered a key result, then its statistical robustness may need some additional support.

Line 295 :La Nina winters….

That idea seems to be not sufficiently developed. I think it is ok to document the increase in frequency of polynya making wind events (with some stats) and leave the global context for future research. Unless the idea can be developed better and/or supported through some results in the literature (Tropical/Arctic connections are a whole study area), I'd leave it out.

Line 345… but none as large as 2018

This is left a bit dangling. What does this mean? Why do you think that is? Increase in strong wind frequency or statistical fluke… or you can't tell at this point?

Comments on prior review.

For independence, I didn't read the prior review in advance but since the D in TCD stands for "Discussion", I might as well. I agree with Frank's point that the delineation from existing work could be done better. It shouldn't be too hard to do this (see also above). Changing the analysis time window to include the missed 1986 polynya would of course be great but likely be a lot of work and require a complete redo of pretty much everything. Maybe a compromise is to reference Frank's unpublished analysis here (Is the discussion in TCD citable?), acknowledge the sensitivity of some results to the time window selection, and discuss the effect of the "missed polynya" on the conclusions. In my view, the existence of a 1986 polynya reinforces the conclusion that sea ice thickness change had little to do with the 2018 Polynya but may qualify the conclusion about the increasing frequency in the last decade (though one Polynya in 1986 doesn't necessarily kill this). Just an idea to consider.

Nice work

Axel Schweiger

---

## Author Comment (AC1)

[Reviewer 1]

The paper describes the performance of the fully-coupled Regional Arctic System Model (RASM) with respect to the simulation of polynya events north of Greenland. A 42-year long simulation (1979-2020) is analysed in combination with satellite products and weather station data. Additionally, two ensembles are generated by forcing RASM with output from the Community Earth System Model (CESM) Decadal Prediction Large Ensemble (DPLE) simulations. The two ensembles, initialized in December 1985 and December 2015, are investigated with respect to precondition of winter polynya events. The paper describes a nice application of dynamical downscaling. However, the paper needs major revision mainly because of two main points of criticism:

[R1-A] It is not immediately clear what the added value of this paper in comparison to the papers of Moore et al. (2018) and Ludwig et al. (2019) is. In both of the latter papers sea ice-ocean models (PIOMAS in case of Moore et al. and NAOSIM in case of Ludwig et al.) are used to analyse the polynya event in more detail as possible with observations alone with almost identical findings (e.g. that preconditioning has no effect on the polynya event in 2018). I suggest to revise the manuscript carefully to make clearer the scientific added value of this study.

→ Moore et al. (2018) focused on how the polynya in February 2018 occurred and Ludwig et al. (2019) addressed what processes were involved in it. They both used an ice-ocean model to study it, which means that the models are prescribed with reanalysis or gridded products on every grid cell. On the other hand, RASM is a fully-coupled high resolution regional model, which allows us to further study interactions between ice, ocean, and atmosphere. In addition, we have investigated every winter polynya event since 1979, which additionally includes polynyas in February 2011 and 2017 and December 1986 in this revised manuscript. Moreover, since RASM is a fully-coupled model (where ocean, atmosphere, and sea ice fields are predicted every time step), it allows us to estimate the number of polynyas that would occur under the observed level of climate warming in each simulation in both ensembles, 30 years apart. It is found that simulated polynyas would occur as long as certain atmospheric conditions are met. This implies that an initial condition of SIT (or decline of SIT due to forced climate warming in the study region) is not a critical factor in occurrence of such polynyas north of Greenland.

[R1-B] Unfortunately, winter is defined in this study from January to March excluding December. If December would have been included the authors would have not missed the polynya north of Greenland in December 1986– Moore et al. (2018) missed it as well because they concentrated on February only (see plot below based on own unpublished analyses). Inspection of the wind field in December 1986 north of Greenland in reveals a northward wind anomaly of almost the same strength and duration as in February 2018. However, there is a dramatic difference. While the occurrence of the 2018 polynya coincides with a sudden stratospheric warming (SSW) event a few days earlier (and associated with a strong decrease in the NAO) the 1986 event does not show any SSW nor any strong NAO change.

→ We thank the reviewer for this suggestion and accordingly have expanded our analysis of winter polynyas to include December, which was missed in Moore et al. (2018). Hence, in the revised manuscript, the winter is defined from December to March and thus the December 1986 polynya is added and thoroughly examined in the subsection 4.2.2 and Fig. 9. In addition, we found that satellite SIC was below 90% in some other years: December 1984 and 2002, but they are excluded because dynamic sea ice transport as defined in this study is too small to count them as polynyas (Fig. S5). Moreover, we further investigated RASM-DPLE ensemble simulations including December, which in turn shows that more polynyas were produced when sea ice was thicker in 1985-1995.

RASM hindcast simulations reproduced the polynya in December 1986 as it is observed in satellite measurements (Figs. 9a and 9b) and confirmed it is a latent heat polynya (Fig. 9c). Although the wind in December 1986 was as strong as in February 2018, its duration was shorter (Fig. S6). If the wind in December 1986 was similar to the wind in February 2018, it is expected that the polynya would be comparable to the observed. As the reviewer reported, all the polynyas except one in February 2018 occurred in non-SSW winters, but we found that there was a link with an AO reversal (Figs. 3a and 9e).

Beside two major points of criticism I listed below a number of points the authors are ask to take into account in the revision. The importance of my suggestions is indicated by minor/major in front of each item but follows the order in the manuscript.

[R1-1]. Minor - line 49 'Introduction': Some of the citations given are pretty old and should be replaced by newer publications. One could be https://journals.ametsoc.org/view/journals/clim/34/13/JCLI-D-20-0848.1.xml
→ We added a newer publication, i.e., Ricker et al. (2021), as well.

[R1-2]. Major – line 60: Figure 1 needs some heavy revision. Panel a) is too dark. Panel b) and c) should be shown in a similar projection as a) to make the comparison easier. The rectangle in b) and c) does not compare well will panel a). Obviously RASM is not able to reproduce the large area of open water north of Fram Strait that can be seen in the observations. This should be discussed in the text and reasons for the deficit should be given (certainly shortcomings in the vertical mixing of the ocean model). The SIT from CFRv2 in panel c) is very unrealistic. A brief discussion on the reliability of SIT from CFSv2 is necessary if the plot should be shown.
→ Fig. 1a is replaced with the VIIRS nighttime image on February 25[th], 2018, after brightened, so that we can compare overall open water areas on the same day in the northern Greenland as well as north of Svalbard. Due to the nature of satellite images, we cannot show the whole Arctic. But Fig 1 rather emphasizes how unrealistic sea ice condition is in CFSv2. Even though the RASM simulation relies on the downscaling of CFSv2 atmospheric boundary conditions, RASM sea ice is very well represented, indicating the potential capability of regional climate models used for dynamical downscaling. We also discussed overestimation of sea ice coverage north of Svalbard where basal melting is dominant. This could be due to that ocean heat transport underestimation along the pathway of the West Spitsbergen Current. Please see the first paragraph of the Discussion.

[R1-3]. Minor - line 63 'Introduction': I was very surprised to see no hint to the SSW when Moore et al. is cited. The coincidence of the polynia with the SSW is mentioned later under 'Discussion' but I would prefer to have some statements about the possible connection to the SSW when citing Moore et al. for the first time because this is the strongest message in that paper.

→ As suggested, we added the sentence, introducing SSW observed in February 2018 in Moore et al. (2018).

[R1-4]. Minor but important - line 124: CS2SMOS should be referenced correctly.

→ CS2SMOS data are referenced and acknowledged as suggested.

[R1-5]. Minor – line 177: Fig. 3 is referenced before Fig. 2. Check the order of the plots. It makes the manuscript unnecessary complicated to read.

→ Fig. 2 is already introduced in the last paragraph of introduction.

[R1-6]. Major – line 179: "… to early March is captured well …". I disagree with the statement (see comment below – line 216).

→ This sentence is about near-surface air temperature variability in the RASM simulation which captures it well as shown in Fig. 3.

[R1-7]. Major – line 216: "The RASM's realistic representation of the polynya…". What seems to be realistic is the size of the polynya but not the location which is very disappointing for a downscaling system. Fig. 2 shows very convincing that the polynya is located too far to the west – the largest fraction of the polynya is located in areas where CS2SMOS shows thicknesses of more than about 1m! Reasons for the mislocation of the polynya should be discussed. In Ludwig et al. a sound estimate of the size of the polynya is given (about 600.000 km2 in maximum) but the size of the polynya in RASM is not compared to this number. This should be done! Instead modelled volume growth rates are discussed for which no observation analog exists (CS2SMOS based estimates are definitely to uncertain given the very coarse resolution). In Ludwig et al. thermodynamical growth rates based on simple estimates are given which should be discussed together with the estimates from RASM.

→ We acknowledge that RASM has a smaller polynya and a more westward position than published observations. We also have made this difference clearer in the revised manuscript. When RASM is used for dynamically downscaling, atmospheric forcings are prescribed only along the lateral boundaries and nudged at approximately the 500 hPa level and above. Hence, surface atmospheric forcing is predicted every time step. Although RASM near surface wind fields agree well with the reanalysis, slight discrepancies in wind direction or magnitude near the study region may shift the center of the polynya more westward. At the same time, it should be noted that CS2MOS SIT is a 7-day mean SIT, not daily. Hence, the direct comparison between them is not straightforward.

Ludwig et al (2019) estimated the size of the of polynya (a maximum extent of about 60,000 km$^2$) based on the satellite SIC, although there are large uncertainties between algorithms,

which makes the comparison less straightforward. For example, MODIS SIC could produce a polynya size that is half of the current estimate (or about 30,000 $km^2$). RASM estimated the polynya size based on SIT less than 10 cm, which gave a maximum size of 13,000 $km^2$, but if the open water area is less than 25 cm of SIT, the maximum size becomes 29,400 $km^2$ (half the size of Ludwig et al. (2019)'s estimate). Also, the RASM integrated thermodynamic ice growth (53 $km^3$) is larger than their study (33 $km^3$).

[R1-8]. Minor – line 232: "… removal due to the polynya …" Not the polynya is removing the sea ice but the winds are removing the sea ice and forming the polynya.
→ It is revised as "ice removal during the polynya formation period"

[R1-9]. Minor – line 237: I do not understand what should be learned from whole subsection.
→ This subsection provides the additional information on how RASM southerly-southeasterly winds contribute to sea ice divergence and thus polynya formation using the SOM analysis. Because the RASM simulation is not forced by reanalysis products, we need to make sure that RASM atmospheric winds are well represented during the polynya period (Fig. 6), and they are confirmed by the ERA-Interim reanalysis wind fields (Fig. S2).

[R1-10]. Major – line 260: The whole subsection 4 might need a revision in the light of the strong wind event in December 1986 mentioned above.
→ We added another subsection for the December 1986 polynya and described how it was developed after thorough analysis (Fig. 9; ice melting, wind pattern, anomalous warming with AO index, dynamic ice volume tendency, and turbulent heat flux over the polynya region). Similar to the recent event in 2018, the strong southerly wind was involved (Table 1), but its duration was shorter (Fig. S6). Hence, this suggests that the size of polynya was smaller than the one in February 2018, as the satellite data indicate in terms of mean SIC in the region (Figs. 7 and S4).   Turbulent heat flux was also lower (Table 2).

[R1-11]. Major – line 367 'Discussion': Obviously, the whole subsection needs reformulation after inspection of the December 1986 wind/polynya event.
→ After thorough analysis of satellite SIC in December 1979-2020, as the reviewer pointed out, we included the missing 1986 winter polynya in the revised manuscript. It is described in the new subsection (4.2.2). In addition, we have inspected RASM-DPLE ensemble simulations including December polynyas as well. It turns out that there are more December polynyas in the 1985-initialized runs than in the 2015-initialized ones. Text has been added to the manuscript to address the inclusion of the December polynyas and the additional conclusions that are made with their additions to the study.

---

## Author Comment (AC2)

[Reviewer 2]

While some of the results are not new, they replicate previous findings (Moore et al. 2018, Ludwig et al. 2019). In particular, they use a coupled atmosphere-ice-ocean modelling framework that overcomes some of the limitations of the ice-ocean framework used in Moore et al. 2018 for testing the impact of sea ice thinning. The analysis regarding the increasing frequency of Polynya maker winds seems also new. I think the paper is generally well conceived and executed and makes a significant contribution to the field and I have no requests for major changes.

The paper might perhaps benefit from some reorganization that better separates the strongly supported results (lack of role of thinning) from the more "future research is needed" ones (increase in polynya frequency due to wind changes). But that's a matter of taste (and likely personal bias) and I think the paper has lingered in reviewer space for too long already so I don't think that should be a requirement.

→ We appreciate your comments. We have expanded our analysis to December and the polynya in December 1986 is now examined in the revised manuscript (see also [R1-B]). This latent heat polynya was also produced by southerly winds as strong as the one in February 2018, but its wind duration was relatively shorter. Hence, the polynya in December 1986 is the second largest one. Although the revised manuscript has become a little lengthy, there are no major changes in terms of findings and conclusions.

Details:
[R2-1] Line 80: Figure 1c. The CFS sea ice thickness looks terrible. Since you are using only above 540 mb probably not an issue for you, but maybe worth a note.
→ This is now mentioned in the beginning of the discussion.

[R2-2] Line 123… "the mean SIC was used to detect … when it dropped below 90%" I think this needs to be justified since the selection of this threshold probably affects the number of polynyas you would have detected? Does the 90% reflect some kind statistical threshold of variability or a value in the literature? This definition probably has to remain somewhat arbitrary but the sensitivity of results to this selection should be discussed somewhere in the paper.
→ It is actually the other way around. When polynya events were observed during 2010s (i.e., February 2011, 2017, & 2018), the daily averaged satellite SIC was dropped below 90% over the study region. Hence, we applied this threshold to detect any additional winter polynya events. The statement was revised accordingly.

[R2-3] Line 165. SOMs. Seems to me that the SOM analysis is a bit of an overkill in this context and may add more confusion than explanation. I think simple wind (anomaly) composites for the Polynya events would have done the job. I know, you did the work and hey, ML! but maybe save for SOM other paper?
→ We used SOM mainly to extract spatial patterns instead of using mean or anomaly fields because of a large size of RASM output: 90 days times four 6-hourly output.

[R2-4] Line 240. Looks like Figure 6 is mentioned before Figure 5 (and I don't see a reference to Figure 5). Figure 5 isn't very interesting at this scale anyhow.
→ Figure 5 is mentioned before Figure 6 in the subsection 4.1.2. Also, as the reviewer 1 suggested, Fig. 5 is revised (including color bar) to show the north of Svalbard to discuss the discrepancy of sea ice condition in February 2018; RASM overestimated its cover (see the 2nd paragraph of the discussion).

[R2-5] Figure 6. Please increase the size. This is hard to see for my aging eyes unless this is improved in production. Also, do we really need to see the full Arctic pattern or is a smaller cut out region sufficient to see what is relevant. What is the significance of the result that this temporal evolution goes through a number of SOM patterns? The key part is northward winds in the region, isn't it?
→ Fig. 6 is revised by increasing the size and the study area is zoomed in as well. The key part is to show how sea ice responds (in terms of ice divergence and convergence) to wind direction/intensity in this region.

[R2-6] Fig 7 (S3,S4). Couldn't those be condensed a bit to highlight the years that should be highlighted? What is the grey shading for the standard deviation? I would have expected the standard deviation of the SIC for that day in the full time series. Why is this larger for the "polynya" cases? See also above comment on defining the SIC threshold. I would have thought a reasonable definition would be 1 or 2 sigma of interannual variability (or quartiles or something like that)?
→ We would like to show that daily mean SIC for the region defined in Fig. 4a from the entire satellite records and determine potential polynya days rather than ones that we already know. We found that those polynya days coincide with when the regional mean SIC is below 90%. The standard deviation indicates spatial variability of SIC in each day. For example, if SIC is spatially homogeneous in a given day, then standard deviation is almost zero. On the other hand, when a polynya occurs, the spatial mean SIC drops down and standard deviation goes up. The figure captions of Figs. S3 and S4 are revised for clarity.

[R2-7] Line 275: Polynya periods... defined as.
How does this interact with your previous definition of 90% SIC. Needs some clarification.
→ Text has been added to clarify that the polynya period is based on the RASM simulation when dynamic sea ice loss (i.e., DVT is less than -10 km$^3$/day) for more than 3 days. The criterion stated above (i.e. 90% satellite SIC) was a sea ice condition, regionally averaged, when we found polynyas over the region. The corresponding statement is also revised.

[R2-8] Line 290.. This section could perhaps get a separate heading "What's driving changes in polynya Frequency".
Totally a style thing, but I would lead with testing the hypothesis that is rejected (thinning ice) and corroborates previous results (Moore et al. 2018) and follow with what the more likely hypotheses of "changes in winds". That would also allow a natural transition to a discussion of the "unanswered" question left for further research (why is the wind changing?).

I like Fig 8 but think that if the increased frequency in wind events is considered a key result, then its statistical robustness may need some additional support.

→ As suggested, a new subsection was assigned to this paragraph. However, we introduced the polynya in December 1986 in the following subsection and made a few changes in the beginning of the paragraph because the polynyas in the 2010s are not unique anymore. This paragraph is also revised by adding the reason why we used two ensembles at the end. In addition, the last paragraph of the new section 4.4. is revised to confirm the hypothesis: the role of southerly winds. Finally, we added a comment about uncertainty of causes changes in wind pattern.

[R2-9] Line 295: La Nina winters….

That idea seems to be not sufficiently developed. I think it is ok to document the increase in frequency of polynya making wind events (with some stats) and leave the global context for future research. Unless the idea can be developed better and/or supported through some results in the literature (Tropical/Arctic connections are a whole study area), I'd leave it out.

→ We agree to the idea that it is not sufficient enough to bring large-scale atmospheric connection. Hence it is removed.

[R2-10] Line 345… but none as large as 2018

This is left a bit dangling. What does this mean? Why do you think that is? Increase in strong wind frequency or statistical fluke… or you can't tell at this point?

→ This sentence is removed due to uncertainty.

[R2-11] Comments on prior review.

For independence, I didn't read the prior review in advance but since the D in TCD stands for "Discussion", I might as well. I agree with Frank's point that the delineation from existing work could be done better. It shouldn't be too hard to do this (see also above). Changing the analysis time window to include the missed 1986 polynya would of course be great but likely be a lot of work and require a complete redo of pretty much everything. Maybe a compromise is to reference Frank's unpublished analysis here (Is the discussion in TCD citable?), acknowledge the sensitivity of some results to the time window selection, and discuss the effect of the "missed polynya" on the conclusions. In my view, the existence of a 1986 polynya reinforces the conclusion that sea ice thickness change had little to do with the 2018 Polynya but may qualify the conclusion about the increasing frequency in the last decade (though one Polynya in 1986 doesn't necessarily kill this). Just an idea to consider.

→ As the reviewer 1 suggested, we decided to include December analysis in our analysis of satellite SIC, RASM hindcast, and RASM-DPLE runs. As you already pointed out, the polynya event in December 1986 further supports the idea that sea ice decline plays little role in more frequent occurrence in recent years. There are some changes in the manuscript because the polynyas in 2010s are not "new" any more, but overall structure, findings, and conclusions are basically the same. Interestingly, RASM-DPLE simulations produced more December polynyas in 1985-1995 than in 2015-2025.

---

## Author Response (AR2)

The paper describes the performance of the fully-coupled Regional Arctic System Model (RASM) with respect to the simulation of polynya events north of Greenland. A 42-year long simulation (1979-2020) is analysed in combination with satellite products and weather station data. Additionally, two ensembles are generated by forcing RASM with output from the Community Earth System Model (CESM) Decadal Prediction Large Ensemble (DPLE) simulations. The two ensembles, initialized in December 1985 and December 2015, are investigated with respect to precondition of winter polynya events.

Although the polynya in 1986 is included now in the revision the main part of the paper has not changed much. I am still not satisfied with the revision for several reasons and still think that the paper needs major revisions.

→ We thank the reviewers for taking the time to provide insightful comments that have improved our manuscript. In this revision, we thoroughly reviewed the manuscript and revised it when it is necessary based on the reviewer's comments.

[Main points of criticism]

[1]. It is still not clear to me what the scientific added value of this paper in comparison to Moore et al. (2018) and Ludwig et al. (2019) is. This should already be clearly stated in the abstract. That the polynya in 2018 is caused by mechanical redistribution is already known from Moore et al. (2018) and confirmed by Ludwig et al. (2019).
→ The abstract in the previously reviewed version of this manuscript already describes the unique contributions of this work. In this revision, we have further modified the abstract and summary to emphasize that this study examined all four winter polynya events ever observed from satellites and modeled using RASM, including 'their evolution and causality, in terms of forced versus natural variability'. To address the latter objectively, we analyzed the results from the model ensemble sensitivity studies about the influence of sea ice thickness on winter polynya development. To our knowledge, no such study has been published to date. The two papers referenced by the reviewer focused solely on the 2018 winter polynya and we have properly acknowledged their findings in the text (i.e., Lines #265-267, #280-283, and #549-552).

[2]. The two ensembles generated by RASM are not analyzed in depth. For instance, it is mainly only mentioned that "The frequency of polynya occurrence had no apparent sensitivity to the initial sea ice thickness in the study area pointing to internal variability of atmospheric forcing as a dominant cause of winter polynyas north of Greenland." This is much too general. I have my doubt, that there is any change at all in the statistics of the upper atmosphere (see below) but ANOVA is the technique to use for exactly this kind of problems (https://en.wikipedia.org/wiki/Analysis_of_variance).
→ A detailed analysis of the RASM-DPLE ensemble runs is the focus of a separate study led by one of the co-authors, Dr. Mark Seefeldt, with a manuscript in the final stages for submission. The manuscript by Seefeldt et al. (in prep.) shows that even though the mean climatic near-surface atmospheric climatic state across the 10-year simulations is similar across the 10 RASM-DPLE ensemble members, the impact on the sea ice state is indeed very different. The differences in the sea ice state are a result of the variability in the evolution and sequencing of the individual weather patterns that result in the mean near-surface climatic state. Addressing mean

climatic state statistics of the upper atmosphere only tells the story of the 10-year mean and it does not tell the story of the individual weather systems, and the sequencing of those weather systems that leads to the state of the atmosphere, sea ice, and polynyas.

[Figure]

Attached is a figure from the upcoming Seefeldt et al. manuscript that demonstrates the large role that variability in the evolution of the weather systems has on the sea ice state. The plot shows the sea ice extent in year 10 of the 10 1985 RASM-DPLE ensemble member simulations. Each ensemble member has a different sea ice extent, especially in Greenland and Barents seas during winter, for 1995 despite a similar near-surface mean climatic across the 10 years for the 10 ensemble members. Although not shown, ANOVA exhibits that the upper tropospheric conditions (i.e., geopotential height at 300 hPa as shown in Fig.S2a) are also similar in the 1985 RASM-DPLE ensemble. It is based on results such as are in this figure that we feel confident that the internal variability of atmospheric forcing plays a large role in the cause of winter polynyas north of Greenland.

As the reviewer suggested, we have also examined the atmospheric condition of geopotential height at the upper troposphere (i.e., 300 hPa) in the RASM-DPLE ensembles using the ANOVA test. It is found that the upper troposphere is statistically different between the two ensembles (see Table S1 and Fig. S2c). However, the polynya-causing winds are similar between the two RASM-DPLE ensembles (See also [4] regarding polynya-favorable winds). An additional figure and ANOVA table are included in the supplementary material in the revised manuscript as Table S1 and Fig. S2.

[3]. There are some typos and grammatically curious formulations in the revision that makes me wonder if the revision was done with the necessary care and if one of the native (American-) English speaking people has seen the revision.

→ All the co-authors have contributed to this work and during this revision process, we paid careful attention to typos and grammar mistakes.

[4].
(4-1) Connected to the second point. I am suspicious that the reduced number of polynyas in the second ensemble (16 versus 25) is just an artifact of the metric used (more then 10km3/day outflow of ice volume for at least three days). The mean thickness in the region of the 1985 ensemble is 3.7m and of the 2015 ensemble 2.8m. Because the metric is based on volume outflow one would expect even without any change in the wind statistics in the two ensembles a reduced number of occurrences in the 2015 ensemble, namely 25 * 2.8m/3.7m ~ 19.
→ The polynya in February 2017 was the smallest one detected from the satellite measurements (Fig. 7). Due to the lack of observational data, the threshold to define a polynya was based on the 2017 event simulated in RASM (Table 1). As the reviewer pointed out, the polynya occurrence did not account for sea ice conditions. Hence, in the revised manuscript, the columns called "the relative total sea ice volume removal (%; i.e., total ice removal ÷ (SIT*study area))" are added in Tables 1, 3, and 4, after factoring in the initial SIT. By applying an additional condition on top of the two existing thresholds, a winter polynya is stringently defined when the ice removed needs to be more than 10.8% (based on the smallest 2017 event observed; see Table 1) relative to the initial volume. We found out that 17 polynyas (reduced from 26) occurred in the first ensemble and 16 polynyas (no change) were observed in the second ensemble (Lines #425-429). The manuscript is revised accordingly but this does not affect our major findings and conclusions because a similar number of polynyas occurred under the different SIT regimes. In addition, even if we applied the 2011 polynya condition (13% ice removal), the same number of polynyas was found: 14 polynyas both in the first and second ensembles (see Lines #430-412 in the Section 4.4).

(4-2) As pointed out under (2) it has to be done a fair statistical analysis (ANOVA) to check if there is a significant difference at all between the two examples regarding the winds that cause the polynyas (there will be near surfaces differences, of course, with respect to the energy balance because the mean SIT is different).
→ The test of statistical significance is performed regarding the polynya-favorable winds between the two RASM-DPLE ensembles: 1985 and 2015 shown in Table 3 and 4, respectively. Because it is not certain how the mean wind speed and stress that cause the polynyas are distributed, a non-parametric test, i.e., the Kolmogorov-Smirnov test, is used to determine whether they are from the same continuous distribution (see Lines #417-423 in the Section 4.4). The test does not reject the null hypothesis at the 5% significance level, suggesting that the polynya-causing winds are similar between the two ensembles. In addition, the mean and standard deviation of the wind speed and stress are provided in the revised tables (see Lines #415-417 in the Section 4.4).

[5]. I mentioned already in the first revision that I find the performance of RASM in simulated the 2018 polynya disappointing. Downscaling is expected to add details to coarse resolution model results but in this case CFSv2 exists in higher resolution (about 0.2 x 0.2 degree) then WRF (about 50km). This should be clearly stated in the manuscript and that 'downscaling' is only done, if at all, with respect to the sea ice and ocean (see e.g. line 162), i.e with respect to the hindcast run the atmospheric variables are rather upscaled then downscaled. I suspect that the relatively coarse WRF resolution is responsible for the location mismatch of the observed and

simulated 2018 polynya (see Fig. 2). It would be nice to give some information about the resolution of CESM-DPLE as well.

→ As stated in the work by Saha et al. (2014) at the beginning of the second paragraph from Section 2, it is stated that the atmospheric model of CFSv2 has a spectral triangular truncation of 126 waves (T126) in the horizontal (equivalent to nearly a 100-km grid resolution) and a finite differencing in the vertical with 64 sigma-pressure hybrid layers (see Section 3.1.1). Hence, RASM-WRF (50 km) is downscaling the 100-km CFSv2 atmosphere for the 2018 winter polynya. It is often mistaken that the resolution of the model output (i.e., 0.2 x 0.2 degree) is considered the model's native resolution (i.e., 100 km). Even if the same resolution was provided by the atmospheric model in RASM, the higher resolution, and Arctic-focused, component models, and higher frequency of coupling with those component models, would provide better results than the reanalysis or global Earth system models. For example, although the RASM hindcast simulation relies on the CFSv2 atmospheric conditions, the winter SIT on 25 February 2018 is very unrealistic in CFSv2 with too thick ice in the central Arctic (Fig. 1c) (see Lines #466-470).

We have revised the model description (Section. 3.1) by adding more information on how the RASM hindcast (Section 3.1.1) and DPLE ensembles (Section 3.1.2) are designed and simulated. We also provided information about the resolution of CFSv2 (Saha et al., 2014) as well as CESM-DPLE, i.e., nominal 1° horizontal resolution and 30 vertical levels (Yeager et al., 2018) (see Lines #165-166 and #185-186, respectively). In addition, we indicated that the mismatch of the polynya location could be improved by increasing the RASM-WRF resolution in the first paragraph of Section 5 (see Lines #462-466).

[6]. There is almost no information given about the coupling of CFSR/v2 or CESM-DPLE. At least some information on the technique, the variables used and the height levels (that information is given in the discussion section, but it should be given in the methods section together with the other information).

→ Here, we assume that the reviewer is asking for more details about how CFSR and CESM data are used to drive RASM. In Section 3.1 of the revised manuscript, more information on how the RASM is set up for fully-coupled simulations (hindcast and DPLE) is given. It is clearly stated that atmosphere forcing data from global earth system models and reanalysis products are derived to provide RASM-WRF with atmospheric initial and lateral boundary conditions. Additionally, it is stated that, due to the limitation in the treatment of the model top boundary layer and stratosphere in RASM-WRF, spectral nudging of winds (i.e., u and v velocities) and temperature is applied linearly for the top half of the model domain, starting approximately above 540 hPa, with a horizontal nudging scale of up to 3400 km as described in Cassano et al. (2017), which also provides more details on information passed between WRF and the coupler in RASM (see Lines #139-199).

[7].
(7-1) Use of the AO: "The daily AO index is constructed by projecting the daily (00Z) 1000mb height anomalies poleward of 20°N onto the loading pattern of the AO" (https://www.cpc.ncep.noaa.gov/products/precip/CWlink/daily_ao_index/ao.shtml) while the loading patterns are derived from monthly mean 1000mb height anomalies. Said that, it is clear that the daily AO is, as well as the monthly AO, a statistical mode that characterizes the Northern hemispheric state of the atmosphere but is not well suited to characterize local atmospheric states

(as e.g. in the polynya region). This is reflected by the low correlation coefficient (e.g. 0.39 and 0.45 in line 313) which means that only about 20% of the variance of the near surface temperature can be 'explained' by (a lagged) AO response (neither the time period used for the calculation of the correlation is given nor it is explained who is leading whom). In other words: About 80% of the variance of the near surface temperature are not correlated to the AO, i.e., independent from it which raises the question why the daily AO is considered at all. The confidence level that is given additionally is not of much help and I wonder if the auto-correlation of the time series is considered or if the daily values are considered as being independent (the auto-correlation might reduce the number of degrees of freedom in the statistical test dramatically).

→ The AO index is defined in Thompson and Wallace (1998) as a climate index of the state of the atmospheric circulation over the Arctic, affecting weather patterns more strongly in some places than in others. Since there were anomalous warming events observed prior to major polynya events in the northern Greenland region, the AO analysis is included to diagnose any potential correlations between the large-scale AO patterns and the small-scale patterns that may lead to the formation of the polynyas. However, the relationships between the observed near-surface temperature and the AO at Station 04301 in 2011 and 2017 are weak and the correlation coefficients are not statically significant when considering a lag-1 auto-correlation. Therefore, the manuscript is revised and states that "near-surface air temperature variability was not statistically correlated with the AO index" in Section 4.2.1 (Lines #311-312) and discussed in Section 5 (Lines #522-524). Also, the polynya incident related to the AO reversal is removed in the Section 6 Summary because the AO is not relevant for understanding polynya formation. If the manuscript did not include an AO analysis, there would be readers that would question why it was not included.

(7-2) In my own analysis of the polynya events 1986 and 2018 I found the plots attached below much more helpful as any correlation coefficients. They show nicely that southerly winds are caused by high pressure systems over the Barents Sea in both events, but that 2018 is connected to an SSW and 1986 not (if the SSW is causing the winds in 2018 or if that is just a coincidence cannot be answered in my opinion but obviously similar strong winds can occur without an SSW event (1986)).

→ In agreement with the reviewer's analysis, the results in the previously reviewed version of this manuscript already emphasized that all four winter polynyas north of Greenland were driven by strong southerly winds in the Sections 4.1.3 (2018 polynya), 4.2.1 (2011 and 2017 polynyas), and 4.3 (1986 polynya) (see also Figs. 4, 6, and 9). We already discussed that SSW was not prerequisite for polynya development because the winter polynyas in 1986, 2011, and 2017 were not associated with SSW (Lines #535-538 in this revised manuscript).

[Minor point]:

[M-1] Section 3.2: Ad hoc it is not clear to me why SOM should be used instead of conventional Empirical Orthogonal Function analysis. A short explanation might be helpful.

→ It is stated that, since atmospheric wind fields might be nonlinear in nature, a linear method, such as the empirical orthogonal function (EOF) analysis, may have drawbacks in extracting nonlinear information (Hsieh, 2004) (see Lines #201-202).

[M-2] Line 235 – 240: Likely the location misfit of the polynya (Fig.3) can be attributed as well to the relatively low resolution of WRF.

→ We have addressed it and discussed that the RASM-WRF resolution needs to be improved for better representation of surface winds (see Lines 462-466).

[M-3] Section 4.1.2: I wonder why 1986 is not mentioned in this section. Are no station data available? If yes, that should be mentioned. If data are available they should be discussed as well for 1986.

→ The polynya that occurred in December 1986 is analyzed and presented in Section 4.3. Also, it is stated that no data are available for the 1986/1987 winter in Section 2.1.

[M-4] Line 250: I find the sentence "Figure 5 shows the RASM thermal sea ice surface, lateral and bottom melting terms were all negligible (< 1 cm) over the study region when integrated for the whole month of February 2018." hard to comprehend. May be better to write " Figure 5 shows that the thermal ice melting terms of RASM (at the surface, lateral, and at the bottom) are all negligible (< 1 cm) over the study region when integrated over February 2018."

→ It is revised as suggested.

[M-5] Line 270-271: "… due to the rapid ice growth during the polynya opening …". The ice is mainly growing after the opening. I suggest: "…  due to the rapid ice growth following the polynya opening …"

→ It is revised as suggested.

[M-6] Line 275-276: "… and have found that the polynya development was associated with strong and persistent winds from the south-southeast." This is no new finding. I suggest: "… and have found that the polynya development was associated with strong and persistent winds from the south-southeast in agreement with Moore et al. (2018) and Ludwig et al. (2019)."

→ It is revised as suggested.

[M-7] Line 314: See main point of criticism (7). If you want to stick to the comparison with the AO: "… lagged by approximately two weeks.": I suggest to make clear that the near surface temperature is leading the AO (which means that the AO can not be causing the temperature anomalies), ie. I suggest "…  while the AO is leading by approximately two weeks.".

→ It is revised as pointed out in the major criticism (please see the response to [7]).

[M-8] Section 4.3: Please check for grammatical correctness. Especially the last sentences "However, the mean turbulent heat flux was much less in December 1986 than in February 2011 even though it was a larger event in terms of polynya size and wind intensity. This is possibly

due to the fact that sea ice was thicker in 1986; for example, the mean SIT was 4.4 m for 5 days before the polynya (Table 1). Due to large open water areas in December 1986, the integrated turbulent heat loss was much larger compared to the polynyas in February 2011 and 2017." are not understandable for me.

→ This section is revised by correcting errors and removing the sentences that are no longer necessary including the relationship with AO because of no observed data (thus, Fig. 9 is revised as well) and the integrated turbulent heat loss.

[M-9] Line 441-455: See major point (4).

→ This paragraph is expanded by adding the null hypothesis test using the Kolmogorov-Smirnov (K-S) two-sample test on the polynya favorable winds. Also, we introduced an additional criterion to define the polynya in RASM-DPLE by taking into account the initial sea ice thickness (please see the response to [4])

[M-10] Line 456: 'longest' → 'largest' ??? or 'longest lasting' ???

→ It is changed to "longest-lasting"

[M-11] Line 493-497: "Overall, the more frequent winter polynyas, produced in a thicker sea ice regime between the two 30-year apart ensembles, implies that changes in SIT are not significant contributors (at least up to now) to the generation of such events for this region during wintertime. Therefore, the findings support that polynyas becomes prevalent when southerly winds are more persistent and stronger in northern Greenland." See my concerns outlined in major point (4). The last sentence I do not understand.

→ The sentences are revised to clarify that the thinning of sea ice is not a significant contributor to the generation of winter polynyas. Even if the initial SIT is considered to define polynyas in RASM-DPLE, the number of polynyas is similar between the two ensembles (please see the response to [4]).

[M-12] Line 500: "… which means that the model is prescribed with reanalysis or gridded products on every grid cell.". That sub sentence is simply untrue. Correct is that forced sea ice-ocean models are one-way coupled. But the reanalysis is not prescribed as stated by the authors. 10m wind is acting via drag formulations on the ice, 2m temperature and humidity act via sensible and latent heat fluxes calculated at the surface by the model and normally downward long- and short-wave fluxes act at the surface but the net fluxes are calculated as well by the model. This sentence should be revised.

→ It is revised as suggested; forced sea ice-ocean models are one-way coupled. Although net heat fluxes are calculated at the surface in the forced ice-ocean model, they are not exchanged across the air-sea interface and do not alter the specified sea ice state.

[M-13] Line 561: The sentence "By taking advantage of an ensemble approach, the internal variability is better assessed with respect to the occurrence of such coastal polynyas during extereme events." needs justification. From which analysis presented I can deduce that with the ensemble approach the internal variability is better assessed? What is meant by 'better'? Better compared to forced sea ice-ocean models? That would be a trivial sentence, of course.

→ This sentence was replaced with the following: 'The combination of decadal dynamical downscaling with an ensemble approach allows us to increase the sample size to 100 winters for each of the two ensembles and thus quantitatively evaluate the impact of decreasing ice thickness on the occurrence of polynyas in the region.'

[M-14] Line 637: I suggest to change the sentence to: "However, the mean turbulent heat loss in the study region during the polynya in 2018 was about 61 W/m 2 (with a maximum of 124 W/m 2 at day XX), which is in good agreement with the results of Ludwig et al. (2019) based on the forced sea ice-ocean model NAOSIM (mean/maximum 40 and 124 W/m 2 , respectively).

→ It is revised as suggested.

[M-15] Line 656: 'that that' -.> 'that'

→ It is corrected.